# Multi-modal generative modeling for joint analysis of single-cell T cell receptor and gene expression data

Felix Drost [1,2], Yang An [1,3], Irene Bonafonte-Pardàs[1], Lisa M. Dratva [4], Rik G. H. Lindeboom [5], Muzlifah Haniffa [4,6], Sarah A. Teichmann [4,7], Fabian Theis [1,2,3], Mohammad Lotfollahi [1,4,8] ✉ & Benjamin Schubert [1,3,8] ✉

Recent advances in single-cell immune profiling have enabled the simultaneous measurement of transcriptome and T cell receptor (TCR) sequences, offering great potential for studying immune responses at the cellular level. However, integrating these diverse modalities across datasets is challenging due to their unique data characteristics and technical variations. Here, to address this, we develop the multimodal generative model mvTCR to fuse modality-specific information across transcriptome and TCR into a shared representation. Our analysis demonstrates the added value of multimodal over unimodal approaches to capture antigen specificity. Notably, we use mvTCR to distinguish T cell subpopulations binding to SARS-CoV-2 antigens from bystander cells. Furthermore, when combined with reference mapping approaches, mvTCR can map newly generated datasets to extensive T cell references, facilitating knowledge transfer. In summary, we envision mvTCR to enable a scalable analysis of multimodal immune profiling data and advance our understanding of immune responses.

T cells are a critical component of the adaptive immune system. Their primary function is the detection of pathogens and tumor cells resulting in immune reactions, which is achieved through antigen recognition by a highly diverse repertoire of T cell receptors (TCRs). While recognizing antigens and immune signaling are well-researched individually, the interplay between T cell function through the TCR and its phenotype remains largely unexplored. Recent findings have shown that T cells sharing the same TCR, so-called clonotypes, express similar transcriptional phenotypes and distribute non-randomly across gene expression-based clusters[1]. Further, differences in memory phenotypes are observed even between T cell clones recognizing the same epitope, suggesting that a layer of T cell transcriptional diversity is clonally inherited[2,3]. These findings indicate that the cells' heritage imprints specific transcriptional cell states shared within clonotypes. Hence, the joint analysis of TCR and transcriptomic information provides a promising tool to identify groups of functionally and clonally linked T cells, thereby improving our understanding of the interdependencies between both modalities.

Paired measurements of TCR and transcriptome can be obtained with modern single-cell multi-omic sequencing techniques[4], enabling the study of cell state and function, simultaneously[5–7]. However, both modalities are usually analyzed separately on the transcriptomic- and TCR levels, potentially missing crucial interdependencies between the two modalities. While integration tools exist for other modality

[1]Computational Health Center, Helmholtz Munich, Ingolstädter Landstraße 1, 85764 Neuherberg, Germany. [2]School of Life Sciences Weihenstephan, Technical University of Munich, Alte Akademie 8, 85354 Freising, Germany. [3]School of Computation, Information and Technology, Technical University of Munich, Boltzmannstraße 3, 85748 Garching bei München, Germany. [4]Wellcome Sanger Institute, Wellcome Genome Campus, Hinxton, Cambridge, UK. [5]The Netherlands Cancer Institute, Plesmanlaan 121, 1066 CX Amsterdam, The Netherlands. [6]Biosciences Institute, Newcastle University, Newcastle upon Tyne NE2 4HH, UK. [7]Department of Physics, Cavendish Laboratory, University of Cambridge, 19 JJ Thomson Avenue, Cambridge, UK. [8]These authors jointly supervised this work: Mohammad Lotfollahi, Benjamin Schubert. ✉e-mail: ml19@sanger.ac.uk; benjamin.schubert@helmholtz-muenchen.de

combinations such as transcriptome and surface protein counts, these methods are not adapted to the TCR protein sequences. While previous tools such as Scirpy[8] and scRepertoire[9] offered utilities for TCR analysis and shared data structures to interlink the unimodal analyses, recent endeavors sought to integrate transcriptomics and TCR information directly. Schattgen et al. used clonotype neighbor graph analysis (*CoNGA*) to detect correlations between TCR sequences and transcriptome[10]. Zhang et al. developed a Bayesian model called TCR functional landscape estimation supervised with scRNA-Seq analysis (*tessa*) to correlate both modalities and cluster T cell clones by their specificity[11]. While these methods incorporate both modalities for clustering, they do not provide an integrated representation for other downstream analyses or offer principled approaches to integrate multiple datasets and scale only to small-size datasets. Furthermore, they use a clonotype-level approach, fusing cells with identical TCRs. However, cells from the same clonotype can have distinct phenotypes[12–14], but this information is lost when reducing cells to common gene expression profiles.

Here, we introduce mvTCR, a multi-view deep learning model for integrating TCR and transcriptome. mvTCR provides a cell-level embedding incorporating both modalities, seamlessly integrates into standard single-cell analysis workflow, and scales well to atlas-level analysis. Moreover, we demonstrate that mvTCR preserves cell state and phenotype information to a high degree, which is crucial for a multimodal analysis. The model provides the contribution of the modalities to its representation for each cell, thereby, providing an additional layer of interpretability. On nine datasets of various sizes, we show that mvTCR's shared representation offers a holistic view of T cells for immunological research, which can be used for various downstream tasks such as antigen-specificity capturing, query-atlas mapping, or the integration of new repertoires. Lastly, we demonstrate the method's ability to reveal SARS-CoV-2-specific and bystander T cells unidentifiable in a unimodal analysis.

## Results

### mvTCR fuses T cell receptor and gene expression data

mvTCR is a generative model based on a deep Variational Autoencoder (Fig. 1, Supplementary Fig. 1, Methods mvTCR) that receives for each cell $i$ the gene expression data $\mathbf{x}^i_{\text{RNA}}$ and the sequence of amino acid IDs of the Complementary Determining Region 3 (CDR3) from the α-chain $\mathbf{x}^i_{\text{TRA}}$ and β-chain $\mathbf{x}^i_{\text{TRB}}$ as the only TCR information. Following[15], we

employed a multi-layer perceptron (MLP) to embed $\mathbf{x}^i_{\text{RNA}}$ to a lower-dimensional representation $\mathbf{h}^i_{\text{RNA}}$. To efficiently capture sequence structure[16], we leverage a transformer network, which learns a contextual representation for each residue by attending to its position and the other amino acids in the CDR3 sequence. This residue-level representation is then aggregated by an MLP to derive a sequence-level representation of the TCR $\mathbf{h}^i_{\text{TCR}}$. After encoding both modalities individually, a mixture module $M$ was used to fuse both modalities into a shared representation $\mathbf{z}^i_{\text{joint}} \sim q(\mathbf{z}^i_{\text{joint}}|\mathbf{h}^i_{\text{RNA}},\mathbf{h}^i_{\text{TCR}})$ for each cell which can be used for various downstream analyses.

We evaluate three approaches to combine the two modalities for $M$: Concatenation, Product-of-Experts (PoE)[17], and Mixture-of-Experts (MoE)[18]. The Concatenation model combines both latent embedding $\mathbf{h}^i_{\text{RNA}}$ and $\mathbf{h}^i_{\text{TCR}}$ as input to an additional encoding network estimating the distribution of $\mathbf{z}^i_{\text{joint}}$. PoE and MoE first estimate separate marginal posterior distributions $q(\mathbf{z}^i_{\text{RNA}}|\mathbf{h}^i_{\text{RNA}})$ and $q(\mathbf{z}^i_{\text{TCR}}|\mathbf{h}^i_{\text{TCR}})$, which are then fused via multiplication or addition to form the joint posterior distribution, respectively. As unimodal baselines, a TCR and a transcriptome model directly estimating the latent distribution from the either $\mathbf{h}^i_{\text{RNA}}$ or $\mathbf{h}^i_{\text{TCR}}$ alone were additionally implemented. To train the model, similar decoding networks reconstructing $\hat{\mathbf{x}}^i_{\text{RNA}}$, $\hat{\mathbf{x}}^i_{\text{TRA}}$, and $\hat{\mathbf{x}}^i_{\text{TRB}}$ were used.

### mvTCR enables analysis of cell function and phenotype

When analyzing T cell repertoires, the antigen specificity of each cell is critical to contextualize its cell state to understand its contribution to the immune reaction. The specificity of T cells towards their cognate epitope is inherently determined by their individual TCR. While TCRs with similar sequences often—but not always—recognize the same epitope[19], dissimilar TCRs may bind the same epitope through distinct binding modes[20], which poses a major challenge for approaches purely based on sequence similarity. Additionally, cells with shared clonal heritage express similar phenotypic characteristics[1–3]. Therefore, we investigated to what extent a joint T cell embedding improves the prediction of antigen specificity while preserving modality-specific variation. To this end, we used a dataset from 10x Genomics with binding annotations $\mathbf{s}^i$ of 44 epitope peptides bound on Major Histocompatibility Complex (pMHC) multimers for four donors (Methods Datasets). The donors greatly varied in T cell specificity. While donors 1 and 2 expressed greater diversity, 68.2% of the T cells of donor 3 were specific to the cytomegalovirus pMHC KLGGALQAK and 26.0% did not

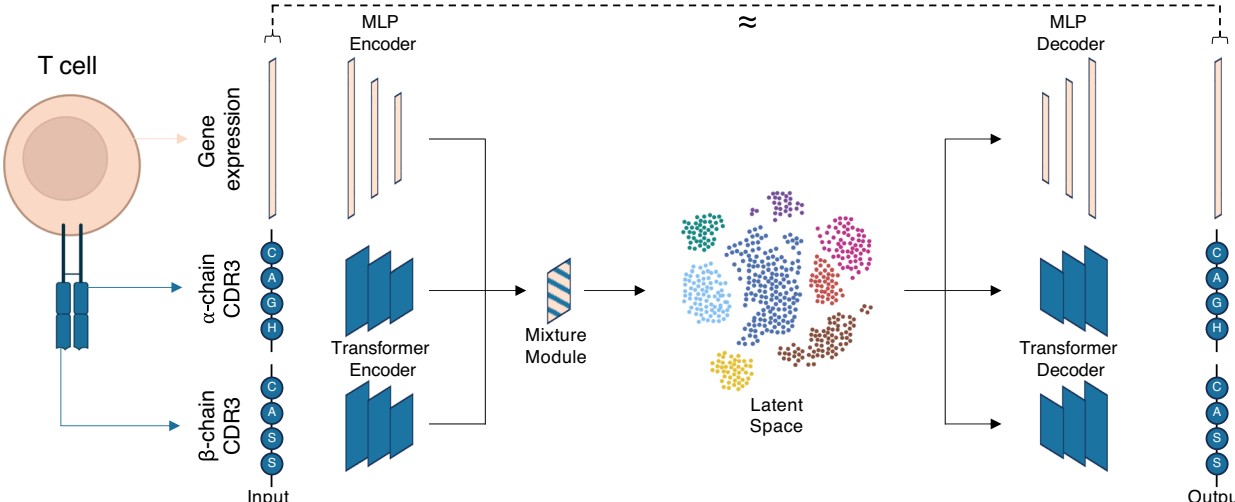

**Fig. 1 | Overview of the mvTCR.** The mvTCR model receives a gene expression vector, along with the CDR3-α and -β amino acid sequences. This input is transformed through separate encoders, and a mixture module combines different modalities to infer a joint representation (latent space) for downstream analysis. This joint representation is then fed into separate decoders to reconstruct the original gene expression and TCR sequences for each cell.

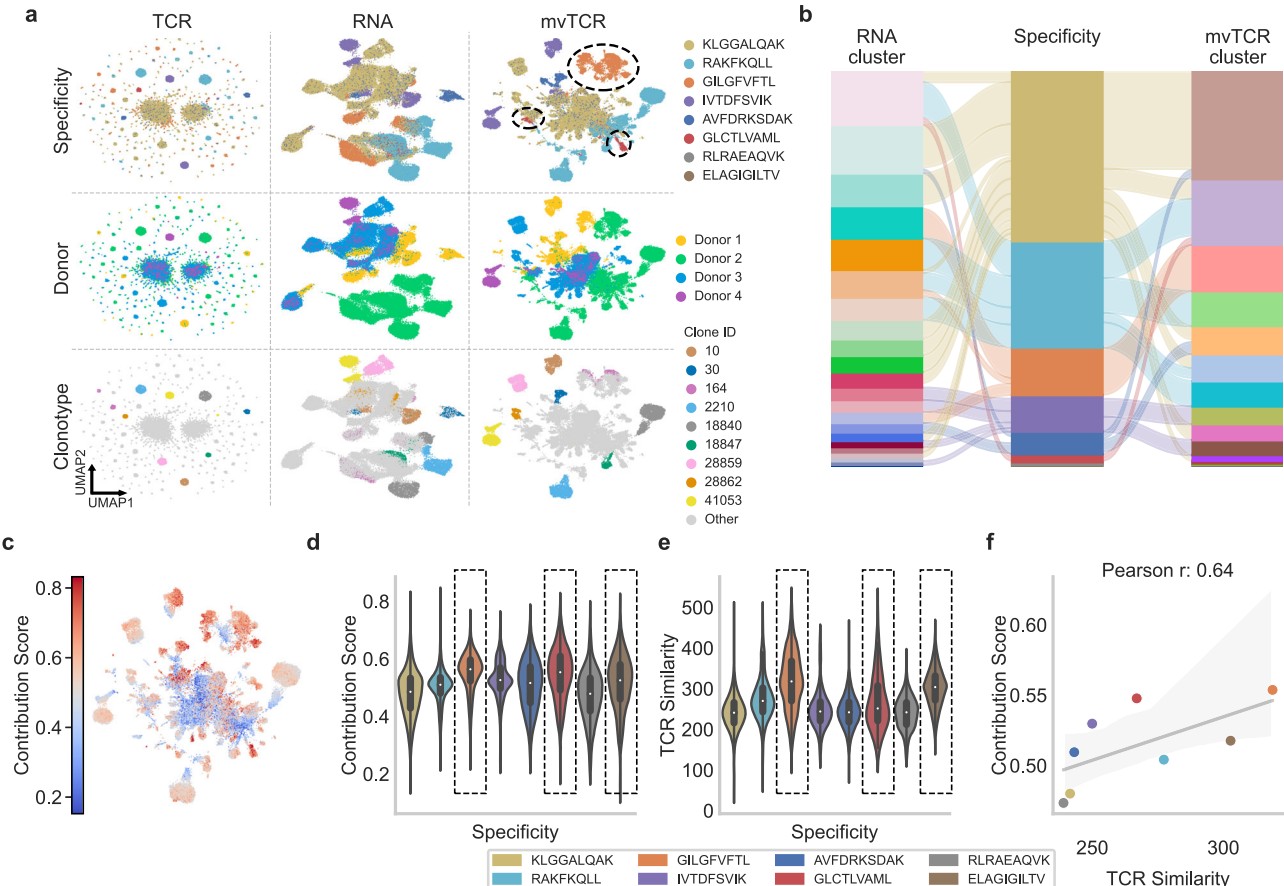

**Fig. 2 | mvTCR learns an interpretable representation of the TCR and transcriptome, highlighting their importance for each cell. a** UMAP visualizations[71] comparing the embeddings of the unimodal (RNA, TCR) and the multimodal mvTCR models colored by peptide-MHC (pMHC) specificity, donor, and ten largest clonotypes. **b** Comparison of RNA clusters to mvTCR clusters with respect to antigen specificity. **c** UMAP visualization colored by the contribution of TCR sequence information on the joint representation. Values <0.5 represent the dominance of RNA information and values >0.5 the dominance of TCR information. **d** Contribution of TCR sequence information to the joint representation by different specificity groups. *n* corresponds to the number of cells in each group (see

Source Data). Selected pMHCs with high TCR Contribution scores are highlighted. **e** TCR similarity by TCRdist[19] between all clonotypes of a specificity group with the pMHCs highlighted as in d. *n* corresponds to the number of unique pairwise clonotype combinations within each group (see Source Data). The box plots within each violin indicate the data quartiles with the whiskers extending to the full distribution excluding outliers outside the 1.5 interquartile range. The median is indicated as a white point. **f** Correlation between the TCR Similarity and the contribution of the TCR information on the mvTCR representation (*n* = 8 specificities). The line indicates the linear regression fit with the 95% confidence interval as error band.

express binding to any of the tested pMHCs. For donor 4, 80.9% of the cells were considered non-binders.

We applied mvTCR to the T cells with specificity towards the eight most common pMHCs (Fig. 2, Supplementary Fig. 2), resulting in 61,237 cells in total. To compare multi- and unimodal representations, the repertoires of all donors were jointly embedded with uni- and multimodal versions of mvTCR. The resulting representations were clustered via the Leiden algorithm[21], where the resolution was chosen to maximize the NMI between pMHC specificity and cluster annotation. The unimodal embedding trained solely on $x_{TCR}$ (Fig. 2a col.: TCR) was dominated by large clonotypes, which form separated clusters of distinct specificities. This was reflected by larger number of clusters (*n* = 214) at a lower Normalized Mutual Information (NMI) of 0.408. Further, the embeddings of different clones did not follow a clear transcriptional pattern to differentiate cell type from each other, which was expected as the TCR does not inherently capture this annotation. In contrast, the transcriptomic model trained on $x_{RNA}$ (Fig. 2a col.: RNA) led to a more continuous representation, which formed several antigen-specific groups with an NMI of 0.456 at *n* = 22 clusters (Supplementary Fig. 3). This performance was influenced by donor-specific biases towards certain epitopes. As GLCTLVAML and RAKFKQLL were bound almost exclusively by Donor 2 (Supplementary Fig. 4), their clusters could be partially identified by shifts in the transcriptomic profile, which shows a clear separation from the remaining dataset. While common T cell markers were distributed across all donors (Supplementary Fig. 5a), Donor 2 differentially expressed several ex vivo activation signature genes such as *FOS*, *DUSP1*, *JUN*, and *NFKBIA*[22] (Supplementary Fig. 5b-d). However, correcting for donor effects using Harmony integration[23] (Supplementary Fig. 6a-b) decreased the clustering performance to an NMI of 0.212 at *n* =14 clusters. In the RNA model, various subpopulations such as T cells binding to GLCTLVAML remained hidden, which were observable in the multimodal mvTCR models (Fig. 2a col.: mvTCR, Supplementary Fig. 2 col.: mvTCR, Concat, PoE). The mvTCR representation was able to combine clonotype and cell type information, simultaneously (Fig. 2a, Supplementary Fig. 2), leading to an NMI of 0.535 (*n* = 15 clusters, Supplementary Fig. 3). While 94.4% of the T cells specific to GILGFVFTL fall into three RNA clusters, 96.5% are contained in only one mvTCR cluster of 99.2% purity (Fig. 2b). Here, T cells from Donors 1 and 2 were combined by the TCR information which were previously separated through inter and intra donor-specific differences in the transcriptomic profile. Similarly, for the pMHC ELAGIGILTV, 86.7% of cells contained in one mvTCR cluster distributed into three RNA clusters, while for IVTDFSVIK 88.6% of the cells group in two mvTCR clusters compared to four RNA clusters. Compared to both unimodal

representations, mvTCR captured antigen-specificity better in this dataset by embedding the T cells into a small number of concise clusters.

To investigate the influence of each modality on the learned representation, we devised a mechanism to infer modality-specific scores (Methods TCR-Contribution). These contribution scores range from complete RNA dominance at 0% to only TCR information at 100% (Fig. 2c) and represent the respective similarity between $z^i_{RNA}$ and $z^i_{TCR}$ to $z^i_{joint}$. The four highest average contributions of the TCR sequences were observed in the pMHCs mentioned above: GILGFVFTL (55.4%), GLCTLVAML (54.8%), IVTDFSVIK (53.0%), and ELAGIGILTV (51.8%). This indicates that the model prioritized TCR information for these cells compared to the cells of the remaining dataset (Fig. 2d). This is corroborated with TCR similarity estimated by the well-established TCRdist method[19], which also indicated an elevated TCR sequence similarity for specificity groups of high TCR-contribution, with RAKFKQLL as the only outlier (Fig. 2e). Further, the average TCR-contribution per specificity and the TCR similarity show a positive Pearson correlation of 0.64 (Fig. 2f).

Next, we evaluated the impact of the transcriptome on the representation. Remarkably, we observed a compact clustering of expanded clonotypes within all three representations persisting even in the RNA-derived model (Fig. 2a), consistent with recent findings suggesting a phenotypic imprint within cells of a clonotype[1–3]. mvTCR preserved the transcriptomic information at 72.0% of the RNA-models' NMI for cell type clustering surpassing the TCR model at 49.5% (Supplementary Fig. 7a). Similarly, clusters defined on RNA coincided at an NMI of 0.684 with mvTCR in contrast to only 0.487 with the TCR model. As the TCR model could solely accumulate cells by their clonotype and homologous sequences, its specificity-based clustering contained only an average of 65.6 different clonotypes, while mvTCR (946.2) groups clonotypes at a similar rate as the RNA-model (728.3) (Supplementary Fig. 7b). The variability within the clonotypes' transcriptomic information and the mvTCR representation showed a strong significant Pearson correlation of 0.802 (p-value < 0.001) independent from their clonal expansion (Supplementary Fig. 7c), which leads to the conservation of cell type and the cells' cytotoxicity (Supplementary Fig. 7d). Within the antigen-specific clusters, cells organize in distinct phenotypic subpopulations, as can be observed by the separation of RNA clusters in the mvTCR embedding. Here, even cells of the same clonotype followed a distinct shift caused by variability in their transcriptome, which underscores the nuanced nature of mvTCR's representation (Supplementary Fig. 7d).

Overall, these results indicate that mvTCR complements its joint embedding with information from both transcriptome and TCR sequences, leading to a more holistic cell representation capturing phenotype and functionality at the same time.

## mvTCR captures antigen-specificity superior to alternative approaches

We then quantitatively evaluated mvTCR's capability to capture antigen specificity. We retrained mvTCR on the 10x dataset pooled over all donors and separately for each donor on five different random splits. Additionally, we included the dataset by Minervina et al.[24] (Minervina dataset), which contains specificity annotation for pMHC-dextramers against 18 SARS-CoV-2 related epitopes for 8,618 cells from 55 vaccinated donors (Methods Datasets). After training the model, a hold-out set was projected onto the training set and the cells' specificity was predicted by a k-Nearest-Neighbor (kNN) model. Note that mvTCR was trained only on clonotypes not contained in the holdout set to ensure unbiased performance estimation. Of the three tested approaches to fuse TCR and GEX information (Methods Network Structure), the MoE performed better on average than the PoE and the Concatenation mixture module (Supplementary Fig. 9), and was therefore used as the main mixing module for the mvTCR model.

To investigate the benefits of mvTCR for predicting specificity, we compared mvTCR against baselines trained only on TCR or transcriptome information (Fig. 3a). On average, mvTCR (F1-Score: 0.821) significantly outperforms both the TCR model (F1-Score: 0.767, p-value: 0.0051, one-sided paired t-test) and the RNA model (F1-Score: 0.759, p-value: 0.0016). In the individual datasets, mvTCR outperforms the unimodal baselines in 7 out of 12 cases during atlas-query prediction while performing on par with the better-performing modality. Often, specificity was either dominated by TCR (donors 1, 2, and 4) or RNA information (Minervina dataset) as shown by better prediction using this modality. To validate that mvTCR focuses on the relevant information, we calculated the contribution of each modality to its representation. We observed a significant Pearson correlation of 0.757 (p-value: 1.3e-6) between the average contribution of TCR information in a dataset to the embedding and the F1-Score quotient of the TCR over the RNA model (Fig. 3b). Depending on the dataset either TCR or transcriptomic information is more indicative of antigen specificity. However, mvTCR adapts to the datasets by focusing on the most informative modality to a larger degree. We found that the average TCR Contribution for the different 10x datasets has a strong negative Pearson correlation with the variation in transcriptome information of large clonotypes (Supplementary Fig. 8). In other words, when cells of the same clonotypes have more diverse gene expression profiles in a dataset mvTCR incorporates less TCR information. As mvTCR was able to fuse both modalities at a task-appropriate weighting for each dataset, it was better suited than unimodal approaches to predict antigen specificity.

Next, we evaluated how good cells with similar antigen specificity cluster in the latent space using NMI as the metric[21] (Fig. 3a, Supplementary Fig. 9). We observed a superior performance of mvTCR when compared to the uni-modal baselines with an increase in NMI of 20.7% over the RNA model (p-value: 1.5e-5) and 7.5% over the TCR model (p-value: 0.048). For clustering, the multimodal representation preserves antigen specificity better than the uni-modal models, except for the TCR model in donor 4. All models fail to capture specificity in donor 3 at an NMI of -0.00. This is caused by the lack of diverse annotation in both datasets with the most prominent binder yielding 74.2% and 92.6%, respectively. Following, we utilized the batch-corrected dataset to ensure that the separation of Donor 2 does not impair the performance in the transcriptome baseline for the pooled 10x dataset. Besides the PCs provided by Harmony[23], we evaluated models trained on the PCs (Harmony-RNA) and additionally the TCR sequences (Harmony-mvTCR). While for the PCs the performance decreased considerably by an F1-Score of 0.239 and an NMI of 0.163 over the RNA model, the models trained on batch-corrected data performed similarly to models trained on the corresponding uncorrected dataset (Supplementary Fig. 10).

To further enhance the representation, we adapted mvTCR to semi-supervised learning introducing a supervised classification head to predict antigen specificity from $\mu^i_{joint}$. While direct prediction fell short, the kNN classifier on the resulting representation surpassed the base model on three datasets, and clustering on all six (Supplementary Fig. 9b) with an average improvement of 0.12 in NMI. These results might further be improved by methods tackling the trade-off between the unsupervised and supervised learning objectives.

Next, we compared mvTCR against tessa for predicting and clustering antigen specificity (Fig. 3c, Methods Benchmarks). For comparability with tessa, which uses the CDR3β sequence as the only input for representing the TCR, we retrained mvTCR without the TCRα sequence. The mvTCR embeddings performed better for atlas-query mapping on all datasets, with an average improvement of 23.6% in F1-Score (p-value: 4.7e-6) for prediction and 27.7% in NMI (p-value: 4.6e-6) for clustering. We attribute this gain in performance to mvTCR's ability to integrate transcriptome information at a cellular level, while such information influences tessa's embedding only on a dataset level through its weighting factors.

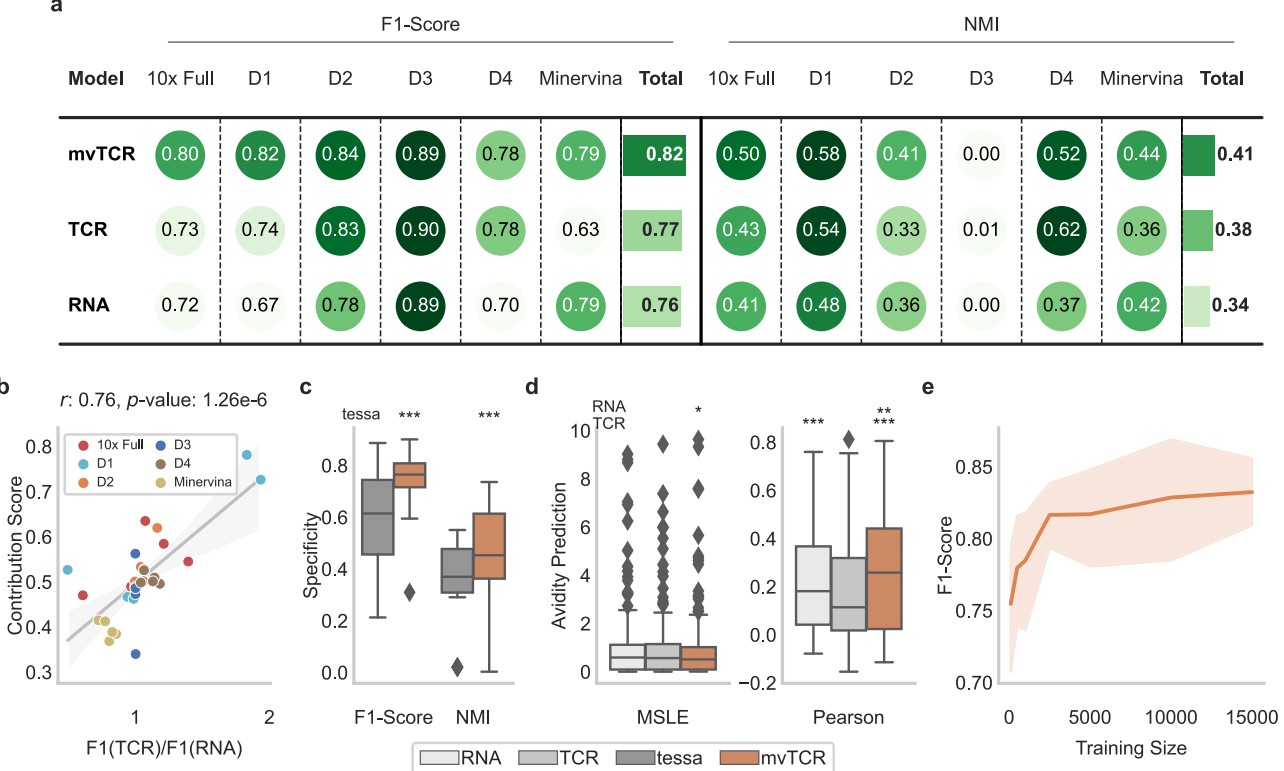

**Fig. 3 | mvTCR's multimodal representation efficiently captures antigen specificity information. a** Predictions of antigen specificity were made on the 10x Genomics dataset for all donors (10x Full), donors 1-4 separately (D1-D4), and the Minervina dataset. Each score represents the average over five random splits ($n = 5$). **b** Correlation between the average TCR-Contribution and the fraction of the F1-Score between the TCR and RNA model for each of the five splits of the six datasets ($n = 30$). $r$ indicates the Pearson correlation coefficient. The line marks the linear regression fit with the 95% confidence interval as error band. **c** Comparison between mvTCR trained only on the gene expression and the CDR3β sequence, and tessa[11] on the tasks defined in a ($p$-values: $p_{F1} = 4.67*10^{-6}$, $p_{NMI} = 4.62*10^{-6}$). **d** Avidity prediction measured by mean squared logarithmic error (MSLE, $p$-values:

$p_{mvTCR-RNA} = 0.0452$) and Pearson correlation ($p$-values: $p_{RNA-TCR} = 6.26*10^{-4}$, $p_{mvTCR-RNA} = 5.95*10^{-3}$, $p_{mvTCR-TCR} = 3.24*10^{-10}$) on each of the five splits and eight specificities of the five versions of the 10x Genomics dataset ($n = 200$). All box plots indicate the data quartiles with the whiskers extending to the full distribution excluding outliers outside the 1.5 interquartile range while the median is indicated as a horizontal line. **e** Influence of mvTCR's training set size on prediction performance at varying dataset sizes ($n_{100-2,500} = 30$, $n_{5,000} = 25$, $n_{10,000} = 15$, $n_{15,000} = 10$). Statistical significance ($p$-values: *<0.05, **<0.01, ***<0.001, baseline indicated left) to the corresponding unimodal representation or the tessa algorithm is calculated via one-sided, paired $t$-test. The bars and lines represent the average metric score, while the error bars and error bands indicate the 95% confidence interval.

We sought to evaluate whether mvTCR can not only predict antigen-specificity but also quantify the binding. As binding avidity depends on the binding strength of the individual TCR as well as the cell state[25], a joint representation might be ideally suited for this task. Therefore, we trained an additional neural network that received the embedding as input to predict the counts of detected for pMHC multimers as an approximate measure of avidity $\mathbf{a}^i$ (Methods Avidity Prediction). The performance greatly varied between datasets, as the prediction was biased toward pMHCs mostly recognized by heavily expanded clonotypes. Again, a multimodal embedding proved beneficial (Fig. 3d, Supplementary Fig. 11a) with a decrease in Mean Squared Logarithmic Error (MSLE) (TCR: 0.045, non-significant; RNA: 0.063, $p$-value: 0.045) and an increase in Pearson correlation (TCR: 0.074, $p$-value: 3.2e-10; RNA: 0.028, $p$-value: 0.006).

In summary, we have demonstrated that the multimodal mvTCR representation effectively encodes antigen specificity information, resulting in superior performance compared to unimodal embeddings and the tessa model for a variety of tasks.

**Cellular heterogeneity is captured at various dataset sizes**
Next, we asked how dataset size influences mvTCR performance. First, we evaluated its robustness for capturing antigen specificity when trained on varying amounts of cells. To this end, we reduced the training set to sample sizes ranging from 100 to 15,000 cells on the 10x Genomics[5] and Minervina dataset[24]. To keep the evaluation

comparable across the different training set sizes, the kNN prediction was performed on the same reference and test data used above (Fig. 3a). We observed that performance improves when the training size increases but begins to saturate at 2500 cells with only a minor increase afterward (Fig. 3e). We, therefore, conclude that mvTCR can be trained already on fairly small datasets containing several thousand T cells.

For single-cell analysis, multimodal embeddings must conserve modality-specific characteristics such as cell type and clonotype. To test this, we trained mvTCR on five different subsamples of three datasets (Methods Datasets) ranging from small study design of 6,713 T cells (Fischer dataset[6]), over a multi-site cohort of 103,761 T cell (Haniffa dataset[7]), to a large-scale collection of multiple studies containing 722,461 T cells (Tumor Infiltrating Lymphocytes (TIL) dataset[26–37]). To provide an estimate of how well modality-specific characteristics are retained, we normalized the NMI to the score of the defining modality (Supplementary Fig. 11b, Supplementary Fig. 12a). The models purely trained on TCR data failed to capture the cell type adequately with an average score of 52.8%. This suggests that the different transcriptomic states are not sufficiently linked to identical or homologous TCR sequences to convey this information. In contrast, mvTCR reaches 83.3% of the RNA models' clustering, a significant improvement compared to the TCR model ($p$-value: 7.5e-6). At the same time, the multimodal representation retained 97.3% of the clonotype information, which is mainly driven by the large clonotype

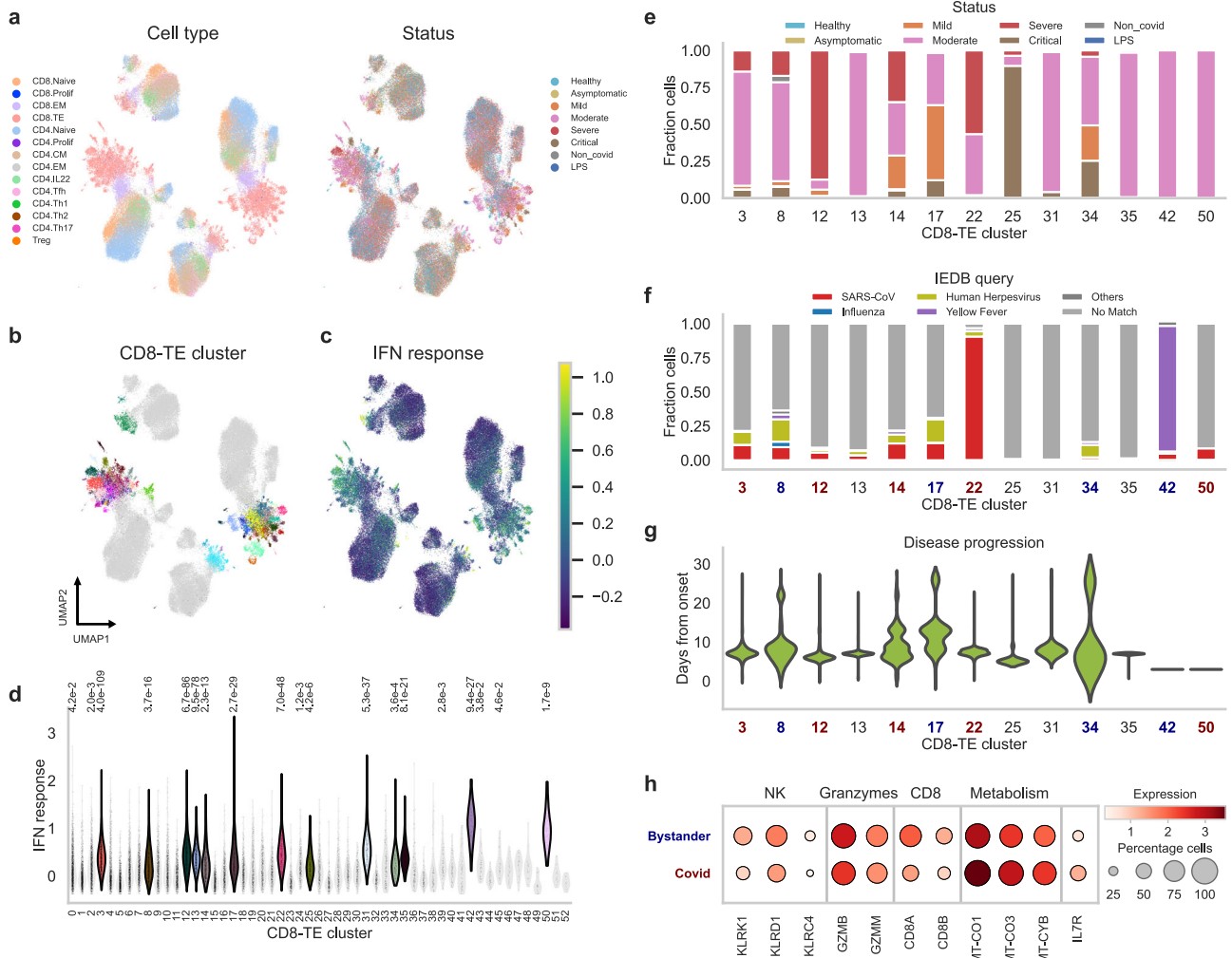

**Fig. 4 | Joint embedding reveals hidden clusters in SARS-CoV-2 study. a** UMAP visualization of the joint embedding of the 103,761 T cells colored by annotation of cell type and status at day of hospital admission. **b** Effector CD8⁺ T cells form separating clusters. **c** UMAP colored by IFN response score, which is elevated in patients with symptomatic SARS-CoV-2 infection. **d** Distribution of the IFN response score across the effector clusters (one-sided, unpaired *t*-test, *p*-values indicated above). The selected, highly significant clusters (*p*-value < 0.001) are marked in color. **e** Patient status of cells from clusters with highly enriched IFN score. **f** Specificity assignment of TCRs within the enriched clusters by query to the IEDB and predicted MHC restriction. **g** Distribution of time after disease onset for the significant clusters. **h** Selected differentially expressed genes between antigen-specific and bystander clusters.

clusters. Interestingly, the RNA model was able to capture this information to 89.8% indicating that many expanded clonotypes follow a similar transcriptomic state. Compared to the tessa model, which preserves TCR information by default as a clonotype level embedding, mvTCR showed an average increase in NMI of 0.182 (*p*-value: 3.1e-7, Supplementary Fig. 12b).

This analysis demonstrated that mvTCR is robust to dataset sizes for capturing antigen-specificity, which makes it applicable to small cohort studies as well as multi-studies atlases. Here, mvTCR simultaneously preserves the characteristics of the cells' TCR to a large degree without sacrificing the transcriptomic information, and vice-versa.

**mvTCR distinguishes activated from bystander T cells**
We asked whether mvTCR can be used to identify the relevant T cell subsets during an immune response to disease. We applied mvTCR on Peripheral blood mononuclear cells (PBMCs) from 130 patients diagnosed with SARS-CoV-2 virus with varying severities, including asymptomatic, mild, moderate, severe, and critical cases[7]. The study contained a negative control group with cells from healthy donors and also patients with other lung diseases, and donors with previously administered intravenous lipopolysaccharide (LPS) to mimic

inflammatory response (Methods Datasets). The resulting mvTCR latent representation (Fig. 4a) separated groups of CD8⁺ effector T cells that expressed a high interferon (IFN) response score[38], which were captured by fine-grained Leiden clustering (Fig. 4b,c). After selecting these clusters of elevated IFN response scores (one-sided, unpaired *t*-test, *p*-values < 0.001, Fig. 4d, Supplementary Data 1), we observed that all 13 resulting clusters contained almost exclusively (99.1%) cells from donors with symptomatic SARS-CoV-2 infection (Fig. 4e). Overall, the selected clusters consisted of cells from samples collected on average after 8.1 ± 4.2 days of symptom onset indicating an ongoing primary T cell response[39,40]. Contrary, the cells of the remaining clusters originated from samples collected at a later date after symptom onset (11.7 ± 8.8 days). Generally, these clusters consisted of several expanded clonotypes with similar TCRs—10 out of 13 clusters had significantly lower inter-clonotype distance[19]—and similar phenotype—12 clusters showed significantly higher inter-cell correlation compared to remaining CD8⁺ effector cells (one-sided *t*-test, *p*-values < 0.05, Supplementary Data 1). Hence, we assumed that cells of a cluster were activated in the same fashion. While several clusters might be SARS-CoV-2 reactive, others might express a bystander response. Bystander T cells are activated by immune signaling without

recognition of their cognate antigen[41] and were reported to play a crucial role in the different severity degrees of COVID-19 patients[42]. To assess T cell specificity, the TCRs were queried to the *Immune Epitope Database (IEDB)*[43], which revealed 1839 cells with possible cognate epitope matches (Methods Datasets), which were reduced to 1342 matches, when filtering out epitopes that were not predicted to bind to the corresponding donors' HLA-types (Supplementary Data 2) via *MHCFlurry 2.0*[44] (Fig. 4f, Supplementary Data 3). Based on this query, we identified five SARS-CoV-2-specific and three bystander clusters. On average, the cells from SARS-CoV-2-specific clusters originated from earlier time points after symptom onset (7.7 days) compared to the bystander clusters (10.1 days) and the remaining clusters (11.6 days) again indicating an antigen-specific T cell response (Fig. 4g).

Differential analysis between antigen-specific and bystander clusters (Fig. 4h, Supplementary Data 4) revealed several upregulated genes related to Natural Killer (NK) cells in the bystander clusters such as *KLRD1*, *NCR3*, and genes of the NK2G receptor group (*KLRK1* and *KLRC4*), which recognize stress-induced self-proteins[41]. Additionally, multiple granzymes (*GZMB*, *GZMM*, and *GZMK*) were upregulated indicating cytotoxic activity. This was coherent with previous studies on bystander activation in various diseases[41,45], which linked elevated levels of *KLRK1* (*NKG2D*) and *NCR3* (*NKp30*) to an Interleukin 15 (*IL-15*) induced T cell response in absence of TCR stimulation. Following *IL-15* exposure, CD8+ T cells adopt an NK-like phenotype and are able to kill targets in an innate-like fashion among others via cytotoxic granzymes. The antigen-specific clusters indicated downregulated CD8 expression (*CD8A*, *CD8B*) as previously reported for an active response of virus-specific CD8+ T cells 8 days after infection[46]. Further, *IL7R* was expressed in 30.3% of the antigen-specific cells. While *IL7R* is downregulated in most CD8+ effector cells, it can be indicative of memory precursor effector cells (MPECs) which survive after viral clearing to form a long-lasting immune memory[47]. Finally, several genes related to the mitochondrial respiratory chain and oxidative phosphorylation (OXPHOS) (*MT-CO1*, *MT-CO2*, *MT-CO3*, *MT-CYB*) were observed in the antigen-specific cells. Even though there is a shift towards aerobic glycolysis after CD8+ activation, a parallel increase in OXPHOS level further contributes to the ATP production[48,49], while both levels are further increased by a peptide-MHC-induced activation[50].

In summary, mvTRC identified clusters of SARS-CoV-2 reactive and bystander cells in a large patient study. The resulting clusters showed compelling concordance of their activation pattern by TCR specificity, time after symptom onset, and differently expressed genes, which demonstrates mvTCR's capability of discovering T cells related on a functional and phenotypical level.

## Multimodal analysis is essential to capture activation mechanisms

We asked if our finding on the SARS-CoV-2 dataset can be replicated by using standard analysis of both modalities without multimodal integration. As an RNA-based method, we clustered the CD8+ effector T cells directly on the transcriptome into a number of clusters similar to the previous analysis using the standard tools provided by Scanpy[51]. As above, the clusters were subsetted based on IFN response score (Supplementary Fig. 13a) and labeled as antigen-reactive or bystander based on the database query (Supplementary Fig. 13c). As a TCR-based analysis, we classified each cell individually based on specificity by directly querying its TCR to the database (Supplementary Fig. 13a). However, the clusters identified with mvTCRs differed greatly from both analyses (Supplementary Fig. 13a,b). Exemplary, the mvTCR SARS-CoV-2 cluster 42, where 88.8% of the cells were annotated as SARS-CoV-2 specificity, was distributed into 37 different RNA clusters. Similarly, the mvTCR bystander cluster 22 consisting of 87.1% yellow fever TCRs, was contained in RNA cluster 20, which also included a similar amount of

SARS-CoV-2 reactive cells and a majority of not annotated cells. The mvTCR clusters lead to an increase in cluster purity of disease severity of 6.7% for RNA-based and 85.9% for TCR-based grouping (Supplementary Fig. 13d) and 16.2% in specificity over the RNA-based clusters (Supplementary Fig. 13c). The mvTCR clusters also showed a higher homogeneity for the time after disease onset with a standard deviation of 2.93 days compared to 3.88 days in RNA clusters and 7.61 in TCR clusters (Supplementary Fig. 14b). Overall, the cells contained in mvTCRs' clusters were in a more similar functional state indicated by phenotype (Supplementary Fig. 14c) and TCR sequences (Supplementary Fig. 14d), which led to a better separation between bystander and antigen-specific cells.

Next, we compared the disparity between the differentially expressed genes (DEGs) of bystander and SARS-CoV-2 reactive cells identified with each of the different approaches (Supplementary Fig. 14e). We observed that out of 95 DEGs based on mvTCR Bystander clusters only 20 overlapped with the 156 DEGs discovered in the RNA-based clusters. Similarly, only 12 DEGs were discovered in both SARS-CoV-2-specific clusters with *IL7R* as the only one from the genes highlighted above (Supplementary Fig. 14f). These numbers are similar to the overlap between mvTCR bystander to RNA-based SARS-CoV-2 clusters ($n = 19$) and vice-versa ($n = 13$). Interestingly, the differential analysis based on the specificity-based annotation resulted in only 16 SARS-CoV-2 and 19 bystander DEGs with no overlap with any of the mvTCR DEGs of that category (Supplementary Fig. 14e-f).

To investigate how mvTCR balances RNA and clonotype information, we selected clones with at least 20 cells. As in the 10x dataset, we observed a significant Pearson correlation of 0.64 (*p*-value < 0.001) between the average within-clonotype RNA and mvTCR distance, independent of their clonal expansion (Supplementary Fig. 15a). Similar to the RNA space, the ten clonotypes with the highest cell type diversity (Supplementary Fig. 15b) were accurately mapped to the effector memory and terminal effector regions of the mvTCR representation (Supplementary Fig. 15c). Conversely, the ten clones purest in cell type were confined to the effector memory region with a lower within-clone distance (Supplementary Fig. 15c). Further, the main transcriptional drivers of T cell variability were well captured by mvTCR as indicated by canonical naive, CD8+ activation, and CD4+ activation markers (Supplementary Fig. 15d).

Overall, we observe how mvTCR enriched transcriptomic information with clonality, allowing us to identify the DEG of clusters which depend both on specificity and transcriptional signatures. This cannot be obtained by the analysis of individual modalities, demonstrating that the integration of multiple modalities using mvTCR enables the discovery of novel knowledge that would have been missed in unimodal approaches.

## mvTCR enables construction of large-scale T cell atlases

Within the domain of single-cell analysis, leveraging reference-based techniques has enhanced the efficiency of rapidly analyzing newly generated datasets[52,53]. However, the effectiveness of such analyses heavily relies on the establishment of robust and comprehensive reference datasets. Often these atlases are composed of multiple smaller-scale studies, where correcting for batch effects is crucial for successful knowledge transfer and analysis. To account for technical effects, we use conditional embeddings (Methods Conditioning). These allow us to construct multi-sample atlases and extend references with query data for knowledge transfer over time and transfer of knowledge from the reference to the query.

We demonstrated this by integrating conditional mvTCR into the scArches framework[52] and integrating data from 12 diverse tumor-infiltrating lymphocyte (TIL) studies[26-37]. After data filtering (Methods Datasets), we had 722,461 T cells across six tissue origins and 11 cancer types. We held out two samples (query) and used the rest to build a reference.

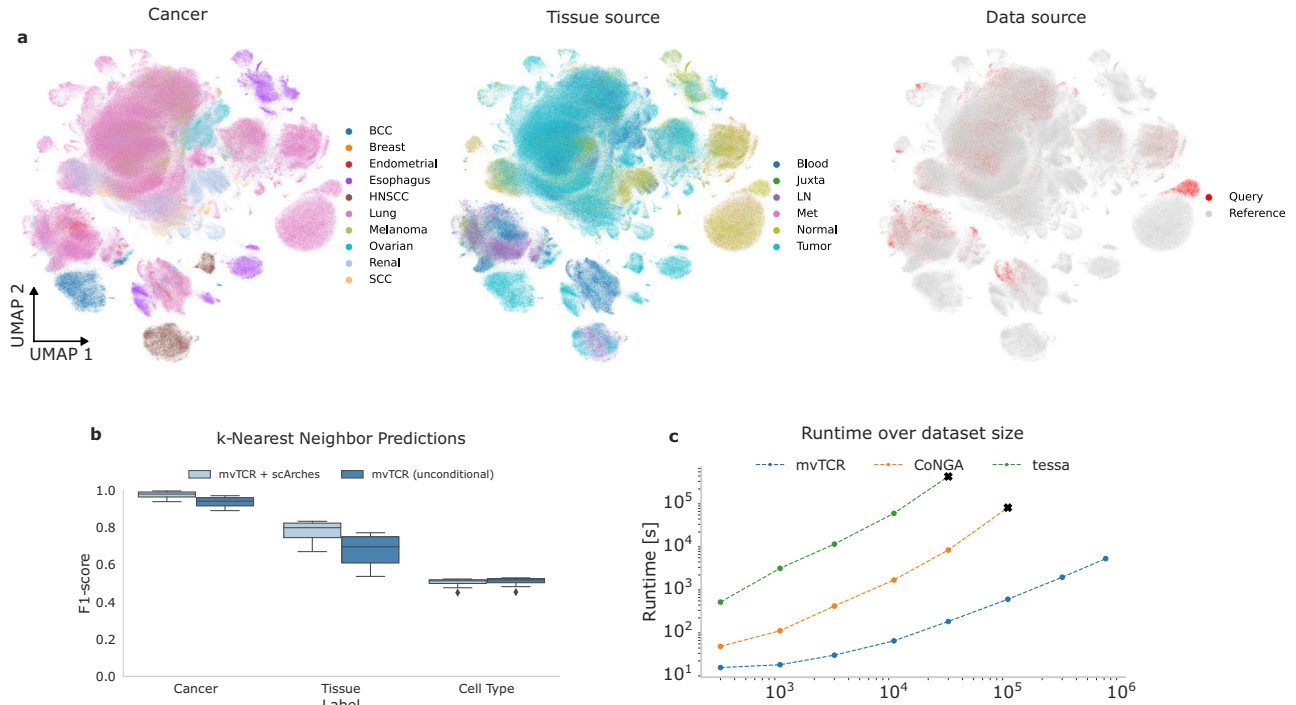

**Fig. 5 | mvTCR enables the construction and querying of multi-modal T cell atlases. a** UMAP visualizations of the joint embedding of reference and query data colored by cancer tissue, tissue type, and data. **b** kNN classification with varying k-values ($n = 10$) using the mvTCR multimodal embeddings as input to classify query cells across different label types. **c** Comparison of model training time for a single trial on different dataset sizes (number of cells) for mvTCR compared against tessa[11] and CoNGA[10]. The box plots indicates the data quartiles with the whiskers extending to the full distribution excluding outliers outside the 1.5 interquartile range while the median is indicated as a horizontal line.

We trained a conditional mvTCR model for reference construction and used scArches for query sample mapping (Fig. 5a, Supplementary Fig. 16a). To evaluate conditional embeddings' effectiveness in batch effect removal and reference mapping, we compared it to a non-conditional method that projected query data into the reference without scArches (Fig. 5b). We assessed their performance in transferring knowledge from the reference to annotate cells for different cancer types, tissue sources, and cell types (Fig. 5a) using a simple kNN classifier with varying k-values (Supplementary Fig. 16b). mvTCR combined with scArches improved tissue type prediction by an average of 4.1% for cancer source and 15.2% while performing similarly for cell type annotations. These results demonstrate the need for technical effect correction and our method's flexibility in integrating into reference mapping approaches. This aligns with recent findings on scRNA-seq data, suggesting that curated, harmonized references enhance contextualizing query datasets[54]. Therefore, we conclude that mvTCR efficiently removes batch effects between query and atlas sets while preserving biological signals across studies.

To further compare the scalability of mvTCR with other established methods integrating gene expression and TCR information, we assessed the execution time for a single training run of mvTCR, CoNGA[10], and tessa[11] on various number of cells obtained by sub-sampling the dataset (Methods Benchmarks). For all dataset sizes, mvTCR was significantly faster than CoNGA and tessa (Fig. 5c). In comparison to CoNGA, mvTCR was up to 135 times faster ($n = 100,000$ cells—mvTCR: 545 s, CoNGA: 73,516 s), while tessa needed up to 2,353-fold more time ($n = 30,000$ cells—mvTCR: 166 s, tessa: 389,578 s). Besides runtime, the memory requirements for CoNGA and tessa were also higher due to pairwise comparisons. On the specified machine used for runtime benchmarking, both CoNGA and tessa exceeded the memory available (256 GB) for dataset sizes of 30,000 and 100,000 cells, respectively, demonstrating mvTCR's great scalability to atlas-scale dataset integration.

## Discussion

With the increasing accumulation of paired single-cell TCR- and RNA-seq datasets, there is a growing demand for scalable methods that can effectively utilize both modalities for integrated analyses. However, standard multimodal methods cannot be applied as they are not adapted to model the TCR sequence. In this context, we introduced mvTCR, a multiview Variational Autoencoder designed to facilitate the large-scale integration of paired TCR and transcriptome data in single-cell studies focusing on T cell repertoires. As the model solely uses the transcriptome counts and the TCR sequences, it is theoretically applicable across species. mvTCR's embedding capability surpasses that of unimodal representations and other multimodal methods, as demonstrated through its efficacy in antigen and avidity prediction, reference mapping, and clustering. One distinctive feature of mvTCR is its ability to provide an additional layer of interpretability for its representation by quantifying the influence of each modality on the representation of individual cells. We observed that the contribution of the TCR sequence correlates with the amount of information it contributes to the formation of antigen-specific clusters. At the same time, the transcriptomic information was captured as demonstrated by the preservation of cell type and state. Even within cells of a clo-notype, mvTCR expressed the variability of the transcriptome leading to nuanced phenotypic patterns in its representation.

mvTCR's representation seamlessly integrates into standard single-cell analysis workflows over the Scanpy-format[51]. Here, its capacity to integrate cell state and function by incorporating both modalities makes it particularly well-suited for identifying clusters of related cells within a repertoire. In a large-scale SARS-CoV-2 dataset, mvTCR effectively distinguished between bystander and antigen-specific T cells. Interestingly, these clusters remained concealed when employing unimodal or step-wise unimodal analysis, underscoring the importance of integrated representations. Lastly, we showcased mvTCR's scalability in handling atlas-level datasets containing several hundred thousand cells for T cell

reference construction. When combined with tools like scArches, mvTCR adeptly mapped new multimodal studies into reference atlases, thereby enabling systematic extensions of these references and the automated analysis of query datasets.

mvTCR has robust performance on multimodal T cell datasets of a variety of biological conditions and dataset sizes. Yet, we applied it mainly to paired data, where gene expression is available in combination with α- and β-CDR3 of the TCR. While training mvTCR on partially missing data might be sufficient for prediction (Supplementary Fig. 17a), novel mosaicing techniques[55,56] could further overcome the decrease in information content. Initial tests suggested that adding surface protein abundance further informed the embedding while still preserving cell type and clonotype (Supplementary Fig. 17b). Therefore, a natural extension would be to incorporate other modalities such as chromatin accessibility as recently proposed by non-TCR-aware single-cell multimodal integration methods[53,57]. Additionally, interpretability methods could be applied to detect TCR and transcriptome characteristics indicative of the cell's functional role. Furthermore, pre-trained TCR embedding models[58] in combination with VDJ-gene encoding could be incorporated besides the CDR3 sequence currently used to improve the TCR representation[59]. Further, mvTCR can be extended to joint B-cell receptor and gene expression datasets. Tests on the SARS-CoV-2 dataset showed that TCR clusters with single amino acid mutations in their CDR3β region have significantly lower distance in the TCR representation of our joint model (Supplementary Fig. 17c). While this indicates that somatic hypermutations of BCRs might be inherently captured by mvTCR, an in-depth benchmark must be conducted upon the availability of large-scale B cell datasets with specificity annotation. As a technical limitation, adjusting the contribution of each modality requires repeated training of mvTCR, ideally by an additional hyperparameter search. Even though an automated selection of network parameters could partially prevent retraining, the desired contribution of each modality is dependent on the study and might vary across or even within one dataset depending on the analysis objective.

In conclusion, we presented mvTCR as a model for analyzing T cell repertoires in the context of infectious disease, tumors, and therapies. We envision that the integration of TCR and transcriptome via mvTCR will uncover hidden interdependencies between the two modalities and identify functionally related T cell sub-clusters, that would remain hidden in a separate analysis of the T cell response, thereby contributing to our understanding of T cell modulation in health and disease.

## Methods

### mvTCR

mvTCR was trained on paired single-cell TCR sequences and RNA-seq datasets. A dataset $\mathcal{D} = \{(\mathbf{x}^i_{TRA}, \mathbf{x}^i_{TRB}, \mathbf{x}^i_{RNA})\}^N_{i=1}$ consists of $\mathbf{x}^i_{TRA}$ and $\mathbf{x}^i_{TRB}$ representing the α- and β-chain of the TCR and $\mathbf{x}^i_{RNA}$ indicating the expression for each cell $i$. $\mathbf{x}^i_{TRA} \in \mathbb{Z}^{seq}$ and $\mathbf{x}^i_{TRB} \in \mathbb{Z}^{seq}$ contain the amino acid sequence of the highly variable CDR3 as the only information on the TCR. Both sequences are tokenized and zero-padded to the maximal sequence length seq present in $\mathcal{D}$. In the following, $\mathbf{x}^i_{TRA}$ and $\mathbf{x}^i_{TRB}$ are summarized as $\mathbf{x}^i_{TCR}$ when both chains are considered. $\mathbf{x}^i_{RNA} \in \mathbb{R}^{genes}$ comprises the 5000 most highly variable genes, whose read counts were normalized and log1p-transformed.

mvTCR encodes the TCR sequences $\mathbf{x}^i_{TRA}$ and $\mathbf{x}^i_{TRB}$ via the two encoder $E_{TRA}$ and $E_{TRB}$, respectively, to obtain the lower-dimensional representations:

$$\mathbf{h}^i_{TRA} = E_{TRA}(\mathbf{x}^i_{TRA}) \text{ and} \tag{1}$$

$$\mathbf{h}^i_{TRA} = E_{TRA}(\mathbf{x}^i_{TRB}) \tag{2}$$

of size $h/2$. Both representations are then concatenated to form the TCR embedding $\mathbf{h}^i_{TCR}$. Similarly, $\mathbf{x}^i_{RNA}$ is transformed via the encoder $E_{RNA}$ to the embedding:

$$\mathbf{h}^i_{RNA} = E_{RNA}(\mathbf{x}^i_{RNA}) \tag{3}$$

of size $h$. Next, both embeddings are combined via different versions of the mixture model $M$ leading to the shared latent distribution:

$$q(\mathbf{z}^i_{joint} | \mathbf{h}^i_{RNA}, \mathbf{h}^i_{TCR}, M) \tag{4}$$

of size $h$. All downstream analysis and benchmark tests were performed on $\mathbf{z}^i_{joint}$. The networks $D_{RNA}$, $E_{TRA}$, and $E_{TRB}$ decode the embeddings to the reconstructions:

$$\hat{\mathbf{x}}^i_{RNA} = D_{RNA}(\mathbf{z}^i_{joint}), \tag{5}$$

$$\hat{\mathbf{x}}^i_{TRA} = D_{TRA}(\mathbf{z}^i_{joint}), \text{ and} \tag{6}$$

$$\hat{\mathbf{x}}^i_{TRB} = D_{TRB}(\mathbf{z}^i_{joint}), \tag{7}$$

### Network structure

mvTCR consists of several networks, specifically, the encoders and decoders for TCR and transcriptome, and different variants of the mixture module for fusing both modalities.

**RNA networks.** Following[15], $E_{RNA}$ uses the architecture of a multi-layer perceptron. Each layer was built by a block containing a fully connected layer, followed by batch-normalization[60], leaky ReLU activation and a dropout layer[61]. Via a linear layer, the output was transformed to $h$. $D_{RNA}$ similarly consisted of these blocks with a final layer with linear activation function obtaining the original input size of 5000 genes.

**TCR networks.** Based on its performance on sequence data in Natural Language Processing, we employed the transformer architecture[62] for extracting features from $\mathbf{x}^i_{TRA}$ and $\mathbf{x}^i_{TRB}$ via the encoders $E_{TRA}$ and $E_{TRB}$. In the transformer, each amino acid token was assigned a trainable embedding which was consequently refined through several multi-head self-attention layers. The output of each encoder was transformed, separately, via a fully connected layer with a linear activation function to $h/2$. While $E_{TRA}$ and $E_{TRB}$ followed the same architecture, they did not share their weights, to allow each network to focus on unique features of their respective input. The shared representation $\mathbf{z}^i_{joint}$ was up-sampled via a fully connected linear layer, before yet again transformer blocks were used for the decoding networks $D_{TRA}$ and $D_{TRB}$. Finally, a linear layer with a softmax activation function reconstructed the amino acid sequence.

For fusing the two modalities, three different versions of the mixture models $M$ were implemented. Additionally, models trained on either the transcriptome or the TCR modality were used as unimodal baseline models.

**Concatenation.** $\mathbf{h}^i_{RNA}$ and $\mathbf{h}^i_{TCR}$ are concatenated to a representation of size $h*2$, which is passed to an additional shared encoding network $E_{joint}$. This network consists of the same blocks described above and estimates the mean $\boldsymbol{\mu}$ and standard deviation $\boldsymbol{\sigma}$ of the normal distribution $q(\mathbf{z}^i_{joint} | \mathbf{h}^i_{RNA}, \mathbf{h}^i_{TCR})$ from which $\mathbf{z}_{joint}$ is sampled via the reparameterization trick[63]. Note, that $\boldsymbol{\mu}$ is used for all downstream analysis throughout this paper.

**Product of Experts (PoE).** Contrary to the concatenation model, PoE uses additional encoder networks $E_1$ and $E_2$ to obtain mean ($\boldsymbol{\mu}_1$ and $\boldsymbol{\mu}_2$)

and standard deviations ($\boldsymbol{\sigma}_1$ and $\boldsymbol{\sigma}_2$) for each modality, individually, resulting in the latent distributions $q(\mathbf{z}^i_{\text{RNA}}|\mathbf{x}^i_{\text{RNA}})$ and $q(\mathbf{z}^i_{\text{TCR}}|\mathbf{x}^i_{\text{TCR}})$. $\mathbf{z}^i_{\text{joint}}$ is sampled via the reparameterization trick from the product of these distributions. $\mu$ and $\sigma$ can be calculated from its closed-form solution

$$q(\mathbf{z}^i_{\text{joint}}|\mathbf{z}^i_{\text{RNA}},\mathbf{z}^i_{\text{TCR}}) = p(\mathbf{z})\prod_m q(\mathbf{z}^i_m|\mathbf{x}^i_m) \; \forall m \in [\text{TCR,RNA}], \quad (8)$$

where $p(\mathbf{z})$ is an univariant Gaussian prior with zero-mean[17]. To motivate the linkage of knowledge between both modalities, the reconstruction was calculated from the shared as well as the modality-specific latent distribution by the same decoder.

**Mixture of Experts (MoE).** As the PoE, MoE calculates individual latent distributions, which are then both used to reconstruct each modality. This forces the encoder networks to have similar predictions for TCR and transcriptomic input[18]. For downstream analysis, the average of both distributions

$$q(\mathbf{z}^i_{\text{joint}}|\mathbf{z}^i_{\text{RNA}},\mathbf{z}^i_{\text{TCR}}) = \frac{1}{2}\sum_m q(\mathbf{z}^i_m|\mathbf{x}^i_m) \; \forall m \in [\text{TCR,RNA}], \quad (9)$$

is used. If not stated otherwise, this mixture module was used throughout this paper.

**Unimodal models.** An encoding network of fully connected blocks described above estimated the mean $\boldsymbol{\mu}$ and standard deviation $\boldsymbol{\sigma}$ of the normal distribution $q(\mathbf{z}^i_{\text{RNA}}|\mathbf{h}^i_{\text{RNA}})$ or $q(\mathbf{z}^i_{\text{TCR}}|\mathbf{h}^i_{\text{TCR}})$ from $\mathbf{h}^i_{\text{RNA}}$ or $\mathbf{h}^i_{\text{TCR}}$, respectively. The reconstruction was calculated from the sampled $\mathbf{z}^i_{\text{RNA}}$ or $\mathbf{z}^i_{\text{TCR}}$.

**Supervised classification head.** The architecture consists of fully connected layers, batch-normalization, ReLU activation, and dropout followed by a final linear layer with softmax activation. The distribution over all specificity labels is predicted from the mean of the joint latent representation $\mu$. The network is trained simultaneously with the VAE to minimize an additional cross-entropy loss, while the weighting of this loss is a tunable hyperparameter.

**TCR-contribution**
To estimate the influence of each modality on each cell embedding, the contribution of input $\mathbf{x}^i_{\text{TCR}}$ and $\mathbf{x}^i_{\text{RNA}}$ to $\mathbf{z}^i_{\text{joint}}$ was determined by first calculating the unimodal hidden representations $\mathbf{h}^i_{\text{TCR}}$ and $\mathbf{h}^i_{\text{RNA}}$. Following, the angular distances

$$\theta^i_m = \frac{\mathbf{z}^i_m \cdot \mathbf{z}^i_{\text{joint}}}{|\mathbf{z}^i_m| * |\mathbf{z}^i_{\text{joint}}|} \; \forall m \in [\text{TCR,RNA}] \quad (10)$$

between each modality and the shared embedding of cell $i$ is determined. The final modality contribution is defined as the difference between $\theta^i_{\text{TCR}}$ and $\theta^i_{\text{RNA}}$ shifted by the factor 0.5 to scale the values between 0 (only RNA contribution) and 1 (only TCR contribution).

**Conditioning**
To integrate query datasets into a trained reference atlas model, we followed a similar approach as Lotfollahi et al.[52]. First, the model is trained on the samples from the atlas datasets to build a reference model. Since the reference atlas may consist of multiple different studies, batch effects can occur between those. To counter batch effects, mvTCR is conditioned toward the studies. Since the MoE mixture module is used in our experiments for query to reference atlas mapping, we describe the procedure for this version of the mixture model only. Let $\mathbf{c}_{j,\text{atlas}}$ be a trainable embedding of dimensionality $D_c$ for each atlas study $j$ representing the difference between studies. For each cell $i$ the corresponding conditional embedding is concatenated

to the hidden representations $\mathbf{h}^i_{\text{RNA}}$ and $\mathbf{h}^i_{\text{TCR}}$ before calculating the individual latent distributions. Similarly, the same embedding $\mathbf{c}_{j,\text{atlas}}$ is concatenated towards the individual latent representations $\mathbf{z}^i_{\text{RNA}}$ and $\mathbf{z}^i_{\text{TCR}}$ before passing them into the corresponding decoders. After the training converged for the reference dataset, the query is integrated using architectural surgery[52]. All parameters of the reference model are frozen and embeddings $\mathbf{c}_{j,\text{query}}$ for the new query studies are randomly initialized. Only these embeddings $\mathbf{c}_{j,\text{query}}$ are trained on the new query datasets. This procedure reduces the number of parameters to be trained by multiple orders of magnitudes while preventing catastrophic forgetting.

**Training**
The models are trained on the weighted sum of reconstruction losses encouraging the conservation of the input data and regularization losses, which shaped the properties of the latent distribution.

$$\mathcal{L}_{\text{total}} = \mathcal{L}_{\text{RNA}}(\mathbf{x}^i_{\text{RNA}},\hat{\mathbf{x}}^i_{\text{RNA}}) + \lambda_1\mathcal{L}_{\text{TCR}}(\mathbf{x}^i_{\text{TCR}},\hat{\mathbf{x}}^i_{\text{TCR}}) + \lambda_2\mathcal{L}_{\text{KLD}}, \quad (11)$$

where

$$\mathcal{L}_{\text{RNA}} = \frac{1}{N}\sum_i^N (\mathbf{x}^i_{\text{RNA}} - \hat{\mathbf{x}}^i_{\text{RNA}})^2 \quad (12)$$

is the mean squared error, and

$$\mathcal{L}_{\text{TCR}} = -\frac{1}{N*K*P}\sum_i^N\sum_p^P\sum_k^K \mathbf{x}^{i,p,k}_{\text{TCR}} \log(\hat{\mathbf{x}}^{i,p,k}_{\text{TCR}}) \quad (13)$$

the Cross-Entropy loss over the sequence encodings for each cell $i$ over each amino acid label $k$ per position $P$ in the TCR. The Kullback-Leibler divergence loss

$$\mathcal{L}_{\text{KLD}} = \text{KL}[q(\mathbf{z}|\mathbf{x}_{\text{RNA}},\mathbf{x}_{\text{TCR}})|p(\mathbf{z})] \text{ with } p(\mathbf{z}) = \mathcal{N}(0,1) \quad (14)$$

constrains the latent distribution to resemble a univariant, zero-mean Normal distribution and is applied to all latent distributions of the respective mixture model. The loss is minimized by the ADAM optimizer with the learning rate as a hyperparameter[64]. The datasets are split into different subsets before training the model. The loss function $\mathcal{L}_{\text{total}}$ is reduced by optimizing all subnetworks of the model jointly on the training data until the validation loss stops decreasing for 5 epochs or a maximum of 200 epochs is reached. Since datasets often contained highly expanded clonotypes, the TCR encoder and decoder focus on over-represented sequences. Therefore, we oversample cells with low-frequency TCRs in the training set of the joint and TCR models by sampling with a probability

$$p_{\text{ct}} = \frac{w_{\text{ct}}}{\sum_{j \in \text{CT}} w_j} \text{ with } w_{\text{ct}} = \log\left(\frac{n_{\text{ct}}}{10}+1\right)^{-1} \quad (15)$$

for each clonotype ct from the set of all clonotypes CT.

For benchmarks (Fig. 3), an additional test set of 20% was used to evaluate the performance on unobserved data. Training, validation, and test sets were constructed randomly on a clonotype level, i.e., cells with the same TCR input sequence were exclusively contained in a single subset.

For evaluating atlas-level integration (Fig. 5a–c), two studies[33,34] containing cells from lung cancer patients were held out of the accumulated dataset and 20% of the remaining data was used as a validation set.

To compare the running times with other multimodal methods (Fig. 5c), mvTCR was trained on random subsets. Again 20% of the data was used as a validation set to measure the time for evaluation calculations. Since the model converged after 20 epochs on the full dataset,

this number was held constant over all subsets, i.e., no early stopping was performed.

## Hyperparameter optimization

To select the best model structure, we perform optimization of all hyperparameters of the architecture via *Optuna*[65] 2.10.0. Depending on the information available, different performance metrics are optimized to obtain the best model over different training runs. When cell-level pMHC specificity information is available (10x dataset and Minervina dataset), the models are evaluated by their ability to capture specificity in the embedding. Specifically, the weighted F1-Score for predicting pMHC specificity via a kNN classifier ($k = 5$) is evaluated between the training (atlas) and validation (query) set. For the remaining datasets, the hyperparameters between the model runs are optimized on how well they cluster cell types provided in the original study and clonotypes defined as identical CDR3-α and -β sequence, simultaneously. These annotations serve as a proxy to what degree the model focuses on gene expression and TCR sequence, respectively. As both pieces of information are partially complementary, it is not possible to conserve both of them simultaneously. E.g. a model only preserving TCR information can perfectly predict the clonotype, but will fail to capture the cell type. Therefore, we can determine the influence that each modality has on the cell representation by choosing a suitable weighting factor between both prediction tasks. In practice, the weighted F1-Score is calculated for assigning both annotations for each cell of the validation set by their nearest neighbor. Models on all datasets were optimized over multiple trials with different sets of hyperparameters for 48 GPU hours, except on the TIL dataset, where the training time was increased to 96 GPU hours due to the dataset size. The timing analysis (Fig. 5c) indicates the runtime for a single trial over different dataset sizes. All experiments except the timing analysis were either conducted on a single GPU machine of 32GB of memory or paralyzed to train 4 models simultaneously on a node containing 4 GPUs and 512GB of memory. All results were obtained from the best-performing model on these performance metrics.

## Avidity prediction

A prediction head is fitted to predict the pMHC tetramer read counts $a^i$ of the most abundant eight pMHCs in the 10x dataset. This additional neural network consists of fully connected blocks with an exponential activation layer. Using the mean squared logarithmic error

$$\mathcal{L}_{\mathrm{MSLE}}(\mathbf{a}^i, \hat{\mathbf{a}}^i) = \frac{1}{N} \sum_{i=0}^{N} (\log(\mathbf{a}^i + 1) - \log(\hat{\mathbf{a}}^i + 1))^2 \qquad (16)$$

between the ground truth and the predicted avidity $\hat{\mathbf{a}}^i$, the models are trained with ADAM optimizer and early stopping (patience of 10). The hyperparameters are optimized by Optuna on 100 training runs.

## Benchmarks

We compare the different multi- and unimodal models of mvTCR and tessa with the following metrics:

**F1-Score.** the performance for predicting cell-level labels with a k-nearest neighbor classifier is evaluated by the harmonic mean between precision and recall. To aggregate performance over all labels $L$, the F1-Score

$$F1 = 2 * \sum_{l \in L} \frac{n_l}{N} * \frac{\mathrm{precision} * \mathrm{recall}}{\mathrm{precision} + \mathrm{recall}} \qquad (17)$$

is weighted by the class support. This metric is applied for predicting antigen specificity on the 10x dataset and for cell type, tissue source, and tissue on the TIL dataset.

**Normalized Mutual Information (NMI).** The NMI is used to compare the overlap between clusters $C$ in the shared embedding and cell labels $L$ via

$$\mathrm{NMI}(L, C) = \frac{2 * I(L, C)}{H(L) + H(C)} \qquad (18)$$

which normalizes the mutual information $I(L, C)$ by the entropies $H(L)$ and $H(C)$. To derive clusters in the latent space, Leiden clustering is applied for different resolution factors (0.01, 0.1, 1.0) and the maximal NMI value between labels and annotation is reported. The NMI is reported for evaluating the clustering of antigen specificity in the 10x dataset, cell type, and reactive clonotypes in the Fischer dataset on the observed data to simulate a realistic analysis scenario. On the TIL dataset the cancer type, tissue source, and cell type are used as labels. The best-performing resolution out of (0.01, 0.03, 0.1, 0.3, 1.0, 3.0) is used for each label individually.

**Mean Squared Logarithmic Error (MSLE).** Following Fischer et al.[66], the MSLE as described in Eq. 16 is used to evaluate the prediction of avidity counts in the 10x dataset.

**Graph connectivity score.** The graph connectivity score quantifies how well cells of the same biological label $l \in L$ are connected in the kNN graph on the embedding space. Following Luecken et al[67]., this metric is calculated as

$$\mathrm{GC} = \frac{1}{|L|} \sum_{l \in L} \frac{|\mathrm{LCC(subgraph)}|}{|l|}, \qquad (19)$$

where LCC(subgraph) indicates all cells within the largest connected component of type $l$, $|l|$ the number of cells from type $l$ and $|L|$ the number of labels. The average over all labels is taken and the metric ranges from 0 to 1. A score of 1 indicates that all cells of type $c$ are connected within one kNN graph.

**Adjusted random index.** The ARI compares the overlap of predicted clusters and biological labels. It assesses both correct overlaps and simultaneously counts correct disagreements. Similar to the NMI score, we use Leiden clustering with the following resolutions (0.01, 0.03, 0.1, 0.3, 1.0, 3.0) and retain the resolution with the best ARI on the labels - cancer type, tissue source, and cell type.

**Average silhouette width.** This score measures the average distance in embedding space of one cell to all other cells while distinguishing between cells of the same and different types. Following Luecken et al.[67], the score is normalized to range between 0 and 1, where 1 indicates that cells are well clustered within each type and separated from clusters of other types. Biological signals should be conserved after integration, hence, a score of 1 is desired. Again, on the TIL dataset the cancer type, tissue source, and cell type are used as labels. On the other hand, for batch effect correction, individual studies should be as indistinguishable as the biological variation allows. Therefore, Luecken et al. modified the calculation, so that 1 represents a perfect overlap of batches[67]. In the experiment, each data source is treated as a unique batch.

Benchmarking tests are performed under the following settings:

**10x dataset.** Optuna optimized the hyperparameters for predicting kNN-prediction of antigen specificity. The model is retrained and evaluated on dataset splits on five different seeds during the benchmark tests (Fig. 3) to enable statistical testing. The avidity prediction (Fig. 3d) is conducted on the same training, validation, and testing splits as indicated above.

**Fischer dataset.** as for the 10x dataset, the models are trained five times on different dataset splits (Supplementary Fig. 12). The hyperparameters are adapted via Optuna to preserve cell type and clonotype at a ratio of one-to-one.

**Comparison to tessa.** We evaluate the performance of mvTCR against tessa as a baseline model (Fig. 3e, Supplementary Fig. 12b). However, tessa takes the CDR3β sequence as the sole input of the TCR. Therefore, we retrain mvTCR without the CDR3α sequence for the 10x and Fischer dataset to avoid the advantage of additional information. Here, we directly use $h_{TRB}^i$ as $h_{TCR}^i$ instead of concatenating it with $h_{TRA}^i$. In this setting, we consider clonotypes as cells with identical CDR3β sequences to avoid the same TCR information in the different subsets of the data. The remaining training follows the description above. After applying the tessa algorithm, kNN predictions are evaluated on the resulting weighted TCR embedding. The cluster annotation provided by tessa is evaluated using the NMI-based metrics.

**Query to reference mapping.** Since the MoE model worked best, we compared this variant with and without architectural surgery[52] on the TIL dataset. Both are optimized using Optuna to determine the best hyperparameter sets to preserve cell type and clonotype with a ratio of 10 to 1. The F1-Score for evaluating cell annotation via the kNN classifier was not statistically analyzed (Fig. 5b).

**Runtime vs dataset size.** In this experiment, we compare the runtime of mvTCR with two concurrent methods integrating gene expression and TCR information - tessa[11] and CoNGA[10] on a computer with 2x Intel Xeon Gold 6226 R (in total 32 Cores), 256 GB RAM, and 1 Nvidia Tesla V100. The same subsets of the full TIL dataset are used for all experiments. We first determined the number of training epochs needed for mvTCR to converge on the full dataset and keep this number (20 epochs) constant over all subsets and no early stopping was performed. The runtime corresponds to the training time for 20 epochs without counting the preprocessing and inference time. Similarly, for tessa and CoNGA, we also excluded the preprocessing time. The time for tessa is counted from running the BriseisEncoder and tessa clustering. CoNGA is run as provided in their example script and the runtime as defined by the original authors is logged.

## Datasets

**10x dataset.** The dataset for all four donors was downloaded from 10x Genomics under the section *Application Note—A New Way of Exploring Immunity*. Following[68], we performed quality control using *Scanpy*[51] 1.7.0, which can shortly be described as following: to remove lysed and dying cells, we filtered cells exceeding a fraction of 20% mitochondrial reads. Additionally, we only considered cells within the span of 1000–10,000 read counts with a minimum of 500 genes per cell. Genes reported for <10 cells were removed from the dataset. Doublets were filtered using *Scrublet* 0.2.3 at a threshold of 0.05[69]. The gene expression data were normalized to 10,000 reads per cell, followed by log1p-transformation and the reduction to the 5000 most highly variable genes. Additionally, the specificity annotation suggested in the publication note was added. All cells not expressing a full TCR consisting of one α- and β-chain were removed from the datasets since mvTCR requires paired information. To ensure correct matching between TCR and specificity in our benchmark data, we further removed all cells expressing multiple TCRs. A clonotype ID was assigned grouping cells with identical α- and β-chain. For better quantification during the benchmark studies, this dataset was reduced to T cells with reported binding to the most abundant eight antigens excluding non-binders. Following, the cell type was annotated via the CellTypist package with default parameters on the full gene matrix[70]. The TCR similarity is calculated as the pairwise distance between the CDR3α and CDR3β sequences as defined by TCRdist[19]. To transform

the values to a TCR similarity, the distance was subtracted from the maximal value occurring within the dataset. No statistical analysis was performed between the TCR contribution across the specificity group (Fig. 2d) and between the TCR similarities (Fig. 2e). For analysis on the batch-corrected dataset, we applied Harmony[23] at its default parameters. Models trained on the batch-corrected dataset received the PCs provided by Harmony as the transcriptome input.

**Minervina dataset.** The raw data were obtained from the online repository provided along with the original publication from Minervina et al.[24]. and reduced to the barcodes provided by the authors to exclude carrier cells, non-T cells, and cells of low quality. The cells were normalized for 10,000 reads and log1p-transformed. Following, the 5000 most variable genes were selected. Additionally, the clonotype ID and the donor as conditional labels were assigned.

**Fischer dataset.** The filtered, normalized, and log1p-transformed dataset of Fischer et al.[6] was downloaded from NCBI GEO. Cells with missing α- or β-chain were removed from the dataset. Clonotypes were assigned for the remaining cells. Finally, we selected the 5000 most variable genes for training mvTCR.

**SARS-CoV-2 dataset.** We obtained the SARS-CoV-2 dataset from the Covid19 Cell Atlas under the section *COVID-19 PBMC Ncl-Cambridge-UCL* and manually joined transcriptomic data with TCR information. Quality control, normalization, and log1p-transformation were already performed in the original publication. The original dataset comprised >780,000 peripheral blood mononuclear cells (PBMCs) from 130 patients collected at three different sites. After filtering for complete TCR annotation, 103,761 cells from 90 patients of the 254,104 annotated T cells remained. We reduced the dataset to the 5000 most variable genes, filtered for incomplete TCRs, and assigned the clonotype. The models were selected to equally preserve cell and clonotype. The database query of TCRs to the IEDB (Fig. 4d) was performed via *Scirpy* version 0.11[8] using the Levenshtein distance with threshold 1. For predicting HLA binding affinity to the resulting epitopes, we used MHCFlurry 2.0 with a threshold of 500 nM[44]. The IFN response score was calculated as the mean of the normalized marker genes described in Szabo et al.[38]. CD8+ effector T cell clusters with elevated IFN response were assigned as antigen-specific if >5% of the cluster's cells and >50% of the epitope matches of the cluster stemmed from SARS-CoV-2 variants. Similarly, bystander clusters were defined as exceeding these thresholds with non-Covid related epitopes. The pairwise similarity of gene expression was calculated as the Pearson correlation of the first 50 principal components of transcriptome space. Differential gene expression was calculated between antigen-specific and bystander clusters via a *t*-test with Benjamini-Hochberg correction using scanpy. Only genes with an adjusted *p*-value <5% and a log-fold change >0.25 were reported. For the comparison of mvTCR to unimodal analysis, the cluster detection in RNA was performed via the Leiden algorithm[21] 0.8.4 on the log1p transformed 5000 most variable genes. To evaluate the addition of a third modality to the mvTCR training, we concatenated the processed and normalized antibody-derived tag counts as provided by the authors to the 5000 most variable genes, and conducted the training as described above. To investigate whether somatic hypermutation is potentially captured by mvTCR we defined single-linkage clusters of TCR clones with a Hamming distance in their CDR3β-chain. The average distances were calculated on the joint model's TCR space ($z_{TCR}^i$) within the cluster or a random clone selection of the same size.

**Tumor-Infiltrating Lymphocyte dataset.** The Tumor-Infiltrating Lymphocyte (TIL) dataset consisted of a collection of studies downloaded as described under https://github.com/ncborcherding/utility. Transcriptome and TCRs were manually merged and cells without

complete TCR were filtered. Additionally, annotated doublets and genes in <100 cells were removed. The data were normalized to 10,000 counts per cell, log1p-transformed, reduced to 5000 most variable genes, and annotated with clonotype. After filtering, the dataset contained 722,461 T cells.

## Reporting summary

Further information on research design is available in the Nature Portfolio Reporting Summary linked to this article.

## Data availability

All datasets used in this paper are publicly available. The 10x dataset was accessed from the 10x website under the Section *Application Note - A New Way of Exploring Immunity* [https://www.10xgenomics.com/datasets] (accessed March, 7th, 2021). The Minervina dataset was accessed from Zenodo under the accession code 6231854 [https://doi.org/10.5281/zenodo.6232103]. The SARS-CoV-2 dataset was accessed from the Covid-19 Cell Atlas under the section *COVID-19 PBMC Ncl-Cambridge-UCL* [https://www.covid19cellatlas.org/index.patient.html] (accessed February, 2nd, 2022). The Fischer dataset was accessed from the NCBI GEO under the accession number GSE171037. The samples contained in TIL dataset stem from a collection of studies. A processed version of this data was downloaded as described in https://github.com/ncborcherding/utility (accessed December, 20th, 2021). Source data are provided with this paper.

## Code availability

The software code including tutorials is available at https://github.com/SchubertLab/mvTCR. The code to reproduce the results of this manuscript can be accessed under https://github.com/SchubertLab/mvTCR_reproducibility. All trained models used for this manuscript can be downloaded from Zenodo via https://doi.org/10.5281/zenodo.8112246.

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

## Acknowledgements

This work was supported by the BMBF grant DeepTCR (#031L0290A), by the Deutsche Forschungsgemeinschaft (DFG, German Research Foundation) (Projektnummer 490846870—TRR355/1 TPZ02), by the Helmholtz Association's Initiative and Networking Fund on the HAICORE@FZJ partition, and Helmholtz International Lab "Causal Cell Dynamics" awarded to B.S. M.L. appreciates F.J.T for enabling and supporting him to conduct this research. Y.A., F.D., and I.B.P. are supported by the Helmholtz Association under the joint research school "Munich School for Data Science - MUDS". M.L. and F.D. acknowledge financial support from the Joachim Herz Stiftung. L.M.D is supported by European Union's Horizon 2020 research and innovation program under the Marie Skłodowska-Curie grant agreement No 955321. Figure 1 and Supplementary Fig. 1 were partially created with BioRender.com.

## Author contributions

M.L. and B.S. conceived the project. F.D., Y.A., and I.B.P. performed research, implemented models, and performed analysis. L.D. and R.L. tested the models and assisted with the analysis. M.H., S.A.T., and F.T. provided feedback on the method and the manuscript. M.L. and B.S. supervised the research. All authors wrote the manuscript.

## Funding

## Competing interests

M.L. consults Santa Anna Bio, owns interests in Relation Therapeutics and is a scientific cofounder and part-time employee at AIVIVO. F.T. reports receiving consulting fees from Roche Diagnostics GmbH and Cellarity Inc., and an ownership interest in Cellarity, Inc. Y.A. acknowledges financial support by JURA Bio, Inc. S.A.T is on the advisory board for Cell

Genomics. L.M.D., R.G.H.L. and S.A.T. are inventors on a filed patent that is related to the detection and application of activated T cells. In the past 3 years, S.A.T. has received remuneration for Scientific Advisory Board Membership from Sanofi, GlaxoSmithKline, Foresite Labs and Qiagen. S.A.T. is a co-founder and holds equity in Transition Bio and Ensocell. From 8 January 2024, S.A.T. is a part-time employee of GlaxoSmithKline. The remaining authors declare no competing interests.
