## [Peer Review File · Nature Communications]

Multi-modal generative modeling for joint analysis of single-cell T-cell receptor and gene expression dataREVIEWER COMMENTS

Reviewer #1 (Remarks to the Author):

The work makes a meaningful attempt to integrate single-cell transcriptomic data with TCR sequence and specificity information. Conceptually, this direction is supported by the work of Paul Thomas' team in Nature Biotechnology (<https://pubmed.ncbi.nlm.nih.gov/34426704/>) and recent findings in Cell Reports ([https://www.cell.com/cell-reports/fulltext/S2211-1247\(23\)01262-7](https://www.cell.com/cell-reports/fulltext/S2211-1247(23)01262-7)), which show distinct scRNA-Seq signatures for T cells with different specificities, among other studies.

In this study, the authors successfully integrated TCR sequence-based information into a dimensionality reduction framework, providing a novel way to interpret scRNA-Seq T-cell data.

My main concern relates to the balance between the two types of information utilized. Starting from Figure 2a, it appears that TCR sequence information dominates over transcriptomic information. Tight cell clusters primarily form based on having the same or homologous TCRs, possibly causing valuable functional information to be overlooked. This represents the key conceptual point of the study—how to maintain a balance where we can assess the functional properties of T cells with the same specificity, utilizing TCR sequence information to bring them closer, while ensuring the transcriptomic information is not displaced in dimensionality reduction.

For reference, the authors could compare with Figure 4a in Cell Reports ([https://www.cell.com/cell-reports/fulltext/S2211-1247\(23\)01262-7](https://www.cell.com/cell-reports/fulltext/S2211-1247(23)01262-7)), where T-cell clones with distinct viral specificities are mapped within the "classic" scRNA-Seq UMAP and are clearly assigned to distinct memory subclasses.

To make the analysis informative, we still need a comprehensive view of functional information. Otherwise, the method may be reduced to primarily visualizing homologous TCR clusters in UMAP.

Other key points to address:

1. It remains unclear which features of the TCR sequence were employed in the model, such as V and J segments, CDR3 length, k-mers, amino acid composition, sequence similarity, etc. The ultimate result seems to correlate with similarity-based TCRdist, indicating that some form of homology information was utilized. This needs to be clearly explained in a manner that can be understood by both bioinformaticians and non-bioinformaticians, capturing the main concept.
2. "In contrast, the transcriptomic model trained on xRNA (Fig. 2a col.: RNA) led to a more continuous representation, which formed several antigen-specific groups with an NMI of 0.456 at $n = 22$ clusters (Supplementary Fig. 3)."

Distinct groups identified at the xRNA level could also possibly correspond to individual donors. Since this part of the study involved four donors, it would be informative to have the same panels from Figure 2a color-coded according to the donor.

This is an important point to help readers understand the extent of data integration and any donor-specific biases (or true differences). We suggest that the authors add donor-based color-coding to each UMAP plot used in the study, preferably within the main figures.

3. "cells from the same clonotype can have other phenotypic differentiation [14, 15]"

Reference 15 is probably not appropriate in this context as it describes infected T cell clones as HIV reservoirs, which is a different topic.

4. "with similar TCRs recognizing similar epitopes."

Yes and no. This may be the case for relatively public TCR clonotypes, but multiple TCRs that have non-similar TCR α or TCR β chains may exist, yet recognize the same antigen. Conversely, some homologous TCRs may recognize distinct antigens, as even minor amino acid changes can significantly impact the TCR surface. This is a general challenge in sequence-based analysis, and we are still far from efficiently predicting the 3D surface of the TCR and its cognate pMHC solely from sequence information. Therefore, this presents a clear limitation for any TCR sequence-based approach, as the logic holds only for highly homologous sequence variants and may still be inaccurate in specific cases. We suggest that the authors address this point to provide clarity to the readers

5. The models for datasets that lack specificity information (e.g., the SARS-CoV-2 dataset) were "optimised on how well they preserve cell type and clonotype measured by the consistency of their label in the embedding within the validation set". It is unclear from the current explanation if the authors optimized the model to provide better "cell type" (CD4.Tfh, CD8.TE, etc.) predictions for the cells in the test set. Additionally, could the authors provide a more detailed explanation of what is meant by "clonotype" in this context? We assume that "clonotype" may not simply refer to the combination of CDR3 α and CDR3 β sequences, as those are directly used as input for the algorithm.

6. We kindly request that the authors review the tutorials provided in the GitHub repository (mvTCR/tutorials/03_training_prediction.ipynb and mvTCR/tutorials/02_training_analysis.ipynb) to improve clarity.

As we understand it, the tutorials serve as guides for utilizing mvTCR on data with annotated TCR specificities (10x dataset in tutorial 03) and data with annotated cell types (haniffa_test in tutorial 02).

Both tutorials currently have an identical header, which seems to refer only to the analysis of the haniffa_test dataset with unknown TCR specificities.

"In this notebook, we will explain how to train your model with the objective of dataset analysis. Here, we optimize to conserve multiple annotations at the same time (e.g. clonotype and celltype). You can determine the influence of both modalities (TCR via clonotype, GEX via celltypes) by specifying a weight for annotation. This might require retraining on a couple of weight values for finding a mixture suitable for your analysis."

Additionally, tutorial 03 contains this block:

Defining Optimization parameters

For this analysis, we will optimize to preserve clonotype and celltype (full_clustering). This optimization mode is called 'pseudo_metric'. By specifying the weight, we can choose the weighting between both modalities.

```
params_optimization = {  
'name': 'knn_prediction',  
'prediction_column': 'binding_name',  
}
```

While tutorial 02 contains this block:

Defining Optimization parameters

For this dataset, we will optimize to predict properties via knn-Atlas-Query-Matching within the dataset. This usually leads to nicely balanced embeddings between TCR and GEX. However, it requires the annotation of TCR specificity in the column 'prediction_column'.

```
params_optimization = {  
'name': 'pseudo_metric',  
'prediction_labels':  
{'clonotype': 1,  
'full_clustering': 5}  
}
```

We assume that the blocks of text might have been swapped, while the blocks of code are correct.

7. Regarding Supplementary Figure 7, the figure caption does not fully match the figure itself as it mentions box plots that are not shown.

8. Could you please clarify whether the dataset size in Figure 5c is measured in terms of cell count? If so, it indicates significantly shorter runtimes compared to the training times mentioned by the authors in the study (48-96 hours). Should users consider Figure 5c as a rough estimation of the runtime for their own real data, or should they still expect longer runtimes of 48-96 hours?

Reviewer #2 (Remarks to the Author):

Review of Multi-modal generative modeling for joint analysis of single-cell T-cell receptor and gene expression data

Summary of the Research:

The research paper introduces mvTCR, a multimodal generative model for the joint analysis of single-cell immune profiling data, specifically focusing on the integration of transcriptome and T-cell receptor (TCR) sequence data. The primary goal of mvTCR is to create a shared representation that captures the unique characteristics of each modality for individual single cells, allowing for the construction of comprehensive T-cell atlases. The study demonstrates that mvTCR enhances the analysis of immune responses, antigen specificity prediction, and clustering when compared to unimodal approaches. It also identifies T-cell subpopulations binding to SARS-CoV-2 antigens, distinguishing them from bystander cells. Additionally, mvTCR can map new multimodal datasets to extensive T-cell references, facilitating knowledge transfer for further analysis.

Advantages of the Study:

Multimodal Integration: mvTCR successfully integrates transcriptome and TCR sequence data, providing a holistic view of single-cell immune profiling, which is crucial for understanding immune responses.

Enhanced Antigen Specificity Prediction: The research demonstrates that mvTCR improves antigen specificity prediction, allowing for a better understanding of T-cell responses in different conditions and diseases.

Clustering: mvTCR effectively clusters cells based on T-cell characteristics, which can aid in the identification of functionally related T-cell subpopulations.

Interpretability: The model offers an additional layer of interpretability by quantifying the contribution of each modality to the representation of individual cells, which is important for understanding the importance of each modality.

Scalability: mvTCR is shown to be robust across varying dataset sizes, making it suitable for large-scale studies.

SARS-CoV-2 Specificity: The study highlights the ability of mvTCR to distinguish between SARS-CoV-2-specific T cells and bystander cells, which is not achievable through unimodal analysis.

Disadvantages of the Study:

Data Dependency: mvTCR is limited to paired data, where gene expression is available in combination with TCR CDR3 data. It may not be applicable to datasets lacking this pairing.

Model Complexity: The study employs a complex deep learning model, which may require substantial computational resources and expertise for implementation and training.

Lack of Real-Time Applications: While mvTCR is valuable for retrospective analysis, its utility in real-time applications, such as diagnostics or treatment decision-making, is not discussed in the paper.

Questions:

Can mvTCR be extended to incorporate other modalities, such as chromatin accessibility or surface protein abundance, to further enhance multimodal integration?

How does mvTCR handle datasets with missing or incomplete TCR sequence data for some cells? Does it affect the model's performance?

How does the model handle the inclusion of a supervised component to predict epitope specificity? Can this enhance the model's predictive capabilities?

The paper discusses the limitations of adjusting the contribution of each modality. Are there any strategies or future directions for automating this process and making it more adaptable to different analysis objectives?

Can mvTCR be applied to other immune cell types, such as B cells, and how might the modeling of somatic hypermutations be addressed in such cases?

In real-world applications, how practical is the implementation of mvTCR in terms of computational resources and time required for training and analysis?

Are there any potential applications for mvTCR in clinical settings, such as disease diagnosis or personalized treatment planning?

Can mvTCR be applied to datasets from diverse species, or are there limitations regarding the type of organisms for which it can be effectively utilized?

How does mvTCR compare with other state-of-the-art methods for integrating multimodal single-cell data in terms of performance and computational efficiency?

In the following, the editor's and reviewers' comments appear in *blue*, our point-by-point answers appear in **black**, and passages and figures we have added or altered in the manuscript are highlighted in **red** in the paper. Short additions are directly added to the point-to-point response, while longer sections are referenced by their line numbers in the revised manuscript.

REVIEWER COMMENTS

Reviewer #1 (Remarks to the Author):

The work makes a meaningful attempt to integrate single-cell transcriptomic data with TCR sequence and specificity information. Conceptually, this direction is supported by the work of Paul Thomas' team in Nature Biotechnology (<https://pubmed.ncbi.nlm.nih.gov/34426704/> [pubmed.ncbi.nlm.nih.gov]) and recent findings in Cell Reports (<https://www.cell.com/cell-reports/fulltext/S2211-1247> [[cell.com](https://www.cell.com/)](23)01262-7), which show distinct scRNA-Seq signatures for T cells with different specificities, among other studies.

In this study, the authors successfully integrated TCR sequence-based information into a dimensionality reduction framework, providing a novel way to interpret scRNA-Seq T-cell data.

C1. Main: My main concern relates to the balance between the two types of information utilized. Starting from Figure 2a, it appears that TCR sequence information dominates over transcriptomic information. Tight cell clusters primarily form based on having the same or homologous TCRs, possibly causing valuable functional information to be overlooked. This represents the key conceptual point of the study—how to maintain a balance where we can assess the functional properties of T cells with the same specificity, utilizing TCR sequence information to bring them closer, while ensuring the transcriptomic information is not displaced in dimensionality reduction.

For reference, the authors could compare with Figure 4a in Cell Reports (<https://www.cell.com/cell-reports/fulltext/S2211-1247> [[cell.com](https://www.cell.com/)](23)01262-7), where T-cell clones with distinct viral specificities are mapped within the "classic" scRNA-Seq UMAP and are clearly assigned to distinct memory subclasses.

To make the analysis informative, we still need a comprehensive view of functional information. Otherwise, the method may be reduced to primarily visualizing homologous TCR clusters in UMAP.

The reviewer raises an important point as the cells' functional state needs to be contained in a multimodal representation. Therefore, we included extensive additional analysis on the 10x dataset and the SARS-CoV-2 dataset to showcase that mvTCR goes beyond a clonotype embedding and, indeed, captures phenotypic properties. We additionally clarified the findings in the chapter on modality conservation which show that both modalities are captured well within the mvTCR embedding. In more detail, we show:

- On the 10x dataset, we demonstrate that phenotypic variability is captured within the mvTCR representation even though we observed a strong conserved phenotypic imprint of clonotypes across all models. In short, we quantitatively demonstrate the preservation of cell type and RNA clusters which is not possible by solely fusing homologous TCRs, leading to clusters containing a higher amount of different clonotypes, and a strong significant correlation between the distances within clones between the RNA and the mvTCR model. We underscore this qualitatively, by showing the cells' phenotype is captured by mvTCR as shown by examples of cell type, cytotoxicity, and RNA clusters. Last, we highlight the shift clonotypes undergo within one specificity cluster based on transcriptomic differences. (**Supplementary Fig. 5, lines 161-177, lines 433-436**: “At the same time, the transcriptomic information was captured as demonstrated by the preservation of cell type and state. Even within cells of a clonotype, mvTCR expressed the variability of the transcriptome leading to nuanced phenotypic patterns in its representation.”)

Supplementary Figure 5 | Transcriptomic information is conserved in the joint embedding. **a**, Clustering performance for cell type derived from the transcriptome and RNA clusters defined by the RNA model. **b**, Amount of clonotypes within each cluster. **c**, Correlation between the average distance within a clonotype in the RNA representation compared to the mvTCR model (p-values: * <0.05 , ** <0.01 , *** <0.001). **d**, UMAP visualizations [22] comparing the embeddings of the transcriptomic and the mvTCR models colored by cell type, cytotoxicity-score [42], clusters defined in the RNA model that occur in mvTCR cluster 2, and the cells of the six largest clonotypes of this cluster. (...)

- We clarified that the contribution of both modalities is not only dataset but also analysis-dependent for which we provide the different mechanisms to optimize the

model. E.g. one weighting might lead to good prediction of antigen-specificity while another weighting is better suited to analyze disease progression.

(Supplementary Fig. 6, lines 205-214, lines 472-473: “Even though an automated selection of network parameters could partially prevent retraining, the desired contribution of each modality is dependent on the study and might vary across or even within one dataset depending on the analysis objective.”)

Supplementary Figure 6 | Dataset-specific modality contribution. Correlation in the 10x dataset between the TCR-Contribution averaged per donor and the average transcriptomic distance within large clonotypes consisting of 20 or more cells. (p-values: $* < 0.05$)

- We highlighted the simultaneous preservation of both modalities on three differently-sized datasets based on clonotype and cell type clustering indicating that the model does not collapse to either TCR or GEX only.
(lines 266-276: “The models purely trained on TCR data failed to capture the cell type adequately with an average score of 52.8%. Apparently, the different transcriptomic states are not sufficiently linked to identical or homologous TCR sequences to convey this information. In contrast, mvTCR reaches 83.3% of the RNA models’ clustering which is a significant improvement compared to the TCR model (p-value: $7.5e-6$). At the same time, the multimodal representation contained 97.3% of the clonotype information, which is mainly driven by the large clonotype clusters. Interestingly, the RNA model is able to capture this information to 89.8% indicating that many expanded clonotypes follow a similar transcriptomic state.”,
lines 279-283: “Here, mvTCR simultaneously preserves the characteristics of the cells’ TCR to a large degree without sacrificing the transcriptomic information, and vice-versa.”)
- On the Haniffa dataset, we demonstrate that important transcriptomic signals are captured by mvTCR on a dataset level. **(Supplementary Fig. 12d, lines 375-377: “Further, the main transcriptional drivers of T cell variability were well captured by mvTCR as indicated by canonical naive, CD8⁺ activation, and CD4⁺ activation markers (Supplementary Fig. 12d).”)**
We further show that even though there is a strong transcriptomic imprint within clonotypes, RNA distances are preserved within mvTCRs embedding, and cell-type diverse clonotypes are accurately distributed.

(Supplementary Fig. 12a-c, lines 368-375: “To investigate how mvTCR balances RNA and clonotype information, we selected clones with at least 20 cells. As in the 10x dataset, we observed a significant Pearson correlation of 0.64 (p-value<0.001) between the average within-clonotype RNA and mvTCR distance, independent of their clonal expansion (Supplementary Fig. 12a). Similar to the RNA space, the ten clonotypes with the highest cell type diversity (Supplementary Fig. 12b) were accurately mapped to the effector memory and terminal effector regions of the mvTCR representation (Supplementary Fig. 12c). Conversely, the ten clones purest in cell type were confined to the effector memory region with a lower within-clone distance (Supplementary Fig. 12c).”)

Supplementary Figure 12 | RNA variation is conserved by mvTCR. **a**, Correlation between the within-clonotype distance in the RNA-space (x-axis) and the mvTCR-space (y-axis). Each point is a clone with a minimum of 20 cells and is colored by the logarithm of the number of cells sharing that clone. (p-values: ***<0.001.) **b**, Amount of each clone's cells and its cell-type purity, computed as the frequency of the most abundant cell type. The dashed line indicates clones with less than 20 cells and excluded from the analysis. The 10 clones with the highest and the lowest cell type purity are highlighted. **c**, UMAP representation of the mvTCR and the RNA colored by selected clonotypes and most frequent cell types. **d**, UMAP representation of the mvTCR and the RNA colored by naive and CD4⁺ and CD8⁺ T cells activation markers.

Other key points to address:

C1.1: It remains unclear which features of the TCR sequence were employed in the model, such as V and J segments, CDR3 length, k-mers, amino acid composition, sequence similarity, etc. The ultimate result seems to correlate with similarity-based TCRdist, indicating that some form of homology information was utilized. This needs to be clearly explained in a manner that can be understood by both bioinformaticians and non-bioinformaticians, capturing the main concept.

R1.1: We agree with the reviewer that our manuscript needs to be graspable for bioinformaticians as well as biological and clinical researchers.

- We clarified the textual description of the input data the model receives in the results (**lines 85-93**: “mvTCR is a generative model based on a deep Variational Autoencoder that receives for each cell the gene expression data and the sequence of amino acid IDs of the Complementary Determining Region 3 (CDR3) from the alpha-chain and beta-chain as the only TCR information. Following [17], we employed a multi-layer perceptron (MLP) (...). To efficiently capture sequence structure [18, 19], we leverage a transformer network, which learns a contextual representation for each residue by attending to its position and the other amino acids in the CDR3 sequence. This residue-level representation is then aggregated by an MLP to derive a sequence-level representation of the TCR sequence (...).”) and the method chapter (**lines 520-521**: “X_TRA (...) and X_TRB (...) contain the amino acid sequence of the highly variable CDR3 as the only information on the TCR.”)
- To further highlight this, we adapted **Figure 1** to contain the label CDR3 to the model input and added the incorporation of VDJ-genes in addition to the CDR3 sequence as future work (**lines 459-461**: “Furthermore, pre-trained TCR embedding models in combination with VDJ-gene encoding could be incorporated besides the CDR3 sequence currently used to improve the TCR representation.”).

C1.2: “In contrast, the transcriptomic model trained on xRNA (Fig. 2a col.: RNA) led to a more continuous representation, which formed several antigen-specific groups with an NMI of 0.456 at n = 22 clusters (Supplementary Fig. 3).”

Distinct groups identified at the xRNA level could also possibly correspond to individual donors. Since this part of the study involved four donors, it would be informative to have the same panels from Figure 2a color-coded according to the donor.

This is an important point to help readers understand the extent of data integration and any donor-specific biases (or true differences). We suggest that the authors add donor-based color-coding to each UMAP plot used in the study, preferably within the main figures.

R1.2: We thank the reviewers for highlighting the importance of donor-specific biases. In fact, the T cells' epitope specificity is intertwined with the donor ID probably due to individual disease history.

- In the revised version of the manuscript, we discuss the unproportional distribution of epitope-specific cells across the donors and briefly discuss the role of donor effects through transcriptomic differences (**Supplementary Fig. 4, lines 133-136**: “This performance was influenced by donor-specific biases towards certain epitopes. As

GLCTLVAML and *RAKFKQLL* were bound almost exclusively by Donor 2 (Supplementary Fig. 4), their clusters could be partially identified by shifts in the transcriptomic profile.”).

Supplementary Figure 4 | Specificity groups in the 10x dataset. Fraction of cells that belong to the different donors in the full 10x dataset separated by the different pMHC specificities.

- To clarify the role of donor-specific effects, we included the Umaps color-coded by the donor ID as suggested by the reviewer (**Fig. 2**). We highlight that mvTCR was partially able to overcome these batch effects by incorporating TCR information (**lines 142-144**: “Here, T cells from Donor 1 and 2 were combined by the TCR information which were previously separated through inter and intra donor-specific differences in the transcriptomic profile.”).

Extract of Figure 2 | mvTCR learns an interpretable representation of the TCR and transcriptome, highlighting their importance for each cell. a. UMAP visualizations [22] comparing the embeddings of the unimodal (RNA, TCR) and the multimodal mvTCR models colored by peptide-MHC (pMHC) specificity, donor, and ten largest clonotypes.

C1.3: “cells from the same clonotype can have other phenotypic differentiation [14, 15]”

Reference 15 is probably not appropriate in this context as it describes infected T cell clones as HIV reservoirs, which is a different topic.

R1.3: We thank the reviewers for pointing this out and agree that the source was not appropriate to support our claim.

- We exchanged it with Bouneaud et al (2005) [15], which shows that approximately two-thirds of clones are shared between T central memory and the T effector memory compartments using a mice model intravenously immunized against the H-Y male antigen. Additionally, we now cite Stemberger et al (2007) [16], which further shows that a single antigen-specific precursor T cell can develop into a wide range of phenotypes (i.e. different types of effector and memory T cells). (**lines 70-71:** “However, cells from the same clonotype can have other phenotypic differentiation [14-16] (...)”)

C1.4: “with similar TCRs recognizing similar epitopes.”

Yes and no. This may be the case for relatively public TCR clonotypes, but multiple TCRs that have non-similar TCR α or TCR β chains may exist, yet recognize the same antigen. Conversely, some homologous TCRs may recognize distinct antigens, as even minor amino acid changes can significantly impact the TCR surface. This is a general challenge in sequence-based analysis, and we are still far from efficiently predicting the 3D surface of the TCR and its cognate pMHC solely from sequence information. Therefore, this presents a clear limitation for any TCR sequence-based approach, as the logic holds only for highly homologous sequence variants and may still be inaccurate in specific cases. We suggest that the authors address this point to provide clarity to the readers

R1.4: We thank the reviewer for suggesting this important point.

- We adjusted the corresponding text passage to address this discussion in more detail about how similar or dissimilar TCR sequences might influence binding. (**lines 109-113:** “While TCRs with similar sequences often - but not always - recognize the same epitope [23-24], even dissimilar TCRs may bind through distinct binding modes [25], which poses a major challenge for approaches purely based on sequence similarity.”)

C1.5: The models for datasets that lack specificity information (e.g., the SARS-CoV-2 dataset) were “optimised on how well they preserve cell type and clonotype measured by the consistency of their label in the embedding within the validation set”. It is unclear from the current explanation if the authors optimized the model to provide better “cell type” (CD4.Tfh, CD8.TE, etc.) predictions for the cells in the test set. Additionally, could the authors provide a more detailed explanation of what is meant by “clonotype” in this context? We assume that “clonotype” may not simply refer to the combination of CDR3 α and CDR3 β sequences, as those are directly used as input for the algorithm.

R1.5: We apologize for the unclear description of the technical details.

- We rephrased the corresponding text passage for more clarity. In detail, we highlighted the information used for cell type and clonotype and explained the problem of conserving both annotations at the same time, which is used to optimize the hyperparameters between multiple model runs. (lines 633-645)

C1.6: We kindly request that the authors review the tutorials provided in the GitHub repository (mvTCR/tutorials/03_training_prediction.ipynb and mvTCR/tutorials/02_training_analysis.ipynb) to improve clarity.

As we understand it, the tutorials serve as guides for utilizing mvTCR on data with annotated TCR specificities (10x dataset in tutorial 03) and data with annotated cell types (haniffa_test in tutorial 02).

Both tutorials currently have an identical header, which seems to refer only to the analysis of the haniffa_test dataset with unknown TCR specificities.

“In this notebook, we will explain how to train your model with the objective of dataset analysis. Here, we optimize to conserve multiple annotations at the same time (e.g. clonotype and celltype). You can determine the influence of both modalities (TCR via clonotype, GEX via celltypes) by specifying a weight for annotation. This might require retraining on a couple of weight values for finding a mixture suitable for your analysis.”

Additionally, tutorial 03 contains this block:

Defining Optimization parameters

For this analysis, we will optimize to preserve clonotype and celltype (full_clustering). This optimization mode is called 'pseudo_metric'. By specifying the weight, we can choose the weighting between both modalities.

```
params_optimization = {  
'name': 'knn_prediction',  
'prediction_column': 'binding_name',  
}
```

While tutorial 02 contains this block:

Defining Optimization parameters

For this dataset, we will optimize to predict properties via knn-Atlas-Query-Matching within the dataset. This usually leads to nicely balanced embeddings between TCR and GEX. However, it requires the annotation of TCR specificity in the column 'prediction_column'.

```
params_optimization = {  
'name': 'pseudo_metric',  
'prediction_labels':  
{'clonotype': 1,  
'full_clustering': 5}  
}
```

We assume that the blocks of text might have been swapped, while the blocks of code are correct.

R1.6: We apologize for this mistake and appreciate the reviewers' detailed investigation of our software, as the usability of our computational tools is of great importance to us.

- We exchanged the indicated part of the tutorial to explain the Hyperparameter optimization by predicting pMHC-specificity and corrected the text in the “Defining Optimization parameters” sections.
- We refactored the API of our software package with functionality to directly transform a single-cell dataset to the appropriate form and train mvTCR with two function calls.

C1.7: Regarding Supplementary Figure 7, the figure caption does not fully match the figure itself as it mentions box plots that are not shown.

R1.7: We apologize for these relics of an earlier version of the figure and thank the reviewer for detecting this.

- The mentioned part of the caption was moved accordingly to the correct figure in the revised manuscript (**Supplementary Fig. 10** (previously 7), **Supplementary Fig. 11** (previously 8)).

C1.8: Could you please clarify whether the dataset size in Figure 5c is measured in terms of cell count? If so, it indicates significantly shorter runtimes compared to the training times mentioned by the authors in the study (48-96 hours). Should users consider Figure 5c as a rough estimation of the runtime for their own real data, or should they still expect longer runtimes of 48-96 hours?

R1.8: We have enhanced the clarity of the corresponding text passage to better explain the timing analysis.

- The dataset size mentioned in Figure 5c is determined by the number of cells, while the runtimes represent the duration of a single training run. In contrast, the 48-96 hours correspond to a full hyperparameter optimization (HPO), where the model is trained several times on varying model specifications.

(Caption Figure 5c: “Comparison of model training time for a single trial on different dataset sizes for mvTCR compared against tessa and CoNGA.”,

lines 410-412: “(...), we assessed the execution time for a single training run of mvTCR, CoNGA, and tessa on various number of cells obtained by subsampling the dataset”,

lines 647-650: “Models on all datasets were optimized over multiple trials with different sets of hyperparameters for 48 GPU hours, except on the TIL dataset, where the training time was increased to 96 GPU hours due to the dataset size. The timing analysis (Fig. 5c) indicates the runtime for a single trial over different dataset sizes.”)

Reviewer #2 (Remarks to the Author):

Review of Multi-modal generative modeling for joint analysis of single-cell T-cell receptor and gene expression data

Summary of the Research:

The research paper introduces mvTCR, a multimodal generative model for the joint analysis of single-cell immune profiling data, specifically focusing on the integration of transcriptome and T-cell receptor (TCR) sequence data. The primary goal of mvTCR is to create a shared representation that captures the unique characteristics of each modality for individual single cells, allowing for the construction of comprehensive T-cell atlases. The study demonstrates that mvTCR enhances the analysis of immune responses, antigen specificity prediction, and clustering when compared to unimodal approaches. It also identifies T-cell subpopulations binding to SARS-CoV-2 antigens, distinguishing them from bystander cells. Additionally, mvTCR can map new multimodal datasets to extensive T-cell references, facilitating knowledge transfer for further analysis.

Advantages of the Study:

Multimodal Integration: mvTCR successfully integrates transcriptome and TCR sequence data, providing a holistic view of single-cell immune profiling, which is crucial for understanding immune responses.

Enhanced Antigen Specificity Prediction: The research demonstrates that mvTCR improves antigen specificity prediction, allowing for a better understanding of T-cell responses in different conditions and diseases.

Clustering: mvTCR effectively clusters cells based on T-cell characteristics, which can aid in the identification of functionally related T-cell subpopulations.

Interpretability: The model offers an additional layer of interpretability by quantifying the contribution of each modality to the representation of individual cells, which is important for understanding the importance of each modality.

Scalability: mvTCR is shown to be robust across varying dataset sizes, making it suitable for large-scale studies.

SARS-CoV-2 Specificity: The study highlights the ability of mvTCR to distinguish between SARS-CoV-2-specific T cells and bystander cells, which is not achievable through unimodal analysis.

Disadvantages of the Study:

Data Dependency: mvTCR is limited to paired data, where gene expression is available in combination with TCR CDR3 data. It may not be applicable to datasets lacking this pairing.

Model Complexity: The study employs a complex deep learning model, which may require substantial computational resources and expertise for implementation and training.

Lack of Real-Time Applications: While mvTCR is valuable for retrospective analysis, its utility in real-time applications, such as diagnostics or treatment decision-making, is not discussed in the paper.

Questions:

C2.1: Can mvTCR be extended to incorporate other modalities, such as chromatin accessibility or surface protein abundance, to further enhance multimodal integration?

R2.1: We thank the reviewer for providing this interesting direction. While mvTCR could in theory be extended to other modalities, we were currently bound to combinations of modalities for which single-cell datasets with several thousands of cells exist. To our knowledge, this is only the case for scRNA+scTCR and scRNA+scTCR+scADT datasets. While no larger-scale scATAC+scTCR datasets exist, mvTCR is, in theory, agnostic to replacing the scRNA input with scATAC, and this might be an interesting direction for future work when these datasets become available as described in the discussion.

- To test whether adding a third modality would provide benefits, we concatenated the Antibody Captured in the SARS-CoV-2 dataset and fed them together with the scRNA input through the MLP encoder. The resulting representation showed a better clustering of disease severity while still preserving cell type and clonotype (**Supplementary Fig. 14b, lines 451-453**: “Initial tests suggested that adding surface protein abundance further informed the embedding while still preserving cell type and clonotype (Supplementary Fig. 14b”, **lines 786-788**: “To evaluate the addition of a third modality to the mvTCR training, we concatenated the processed and normalized antibody-derived tag counts as provided by the authors to the 5,000 most variable genes, and conducted the training as described above.”), which indicates that mvTCR is extendable to additional modalities, as well.

Extract of Supplementary Figure 14 | Extensions of mvTCR. b, Clustering of mvTCR with TCR, transcriptome, and surface protein markers compared to the baseline model on the SARS-CoV-2 dataset. (...)

C2.2: How does mvTCR handle datasets with missing or incomplete TCR sequence data for some cells? Does it affect the model's performance?

R2.2: This comment raises an interesting question as filtering for complete information reduces the amount of training and analysis data.

- For a comparable evaluation to our previous experiments, we simulated missing TCR and GEX information on the full 10x dataset by randomly deleting 15% of TCR alpha- and beta-chains as well as zeroing out the GEX data of 15% of the cells. The evaluation was conducted as previously shown in Figure 2. We observe a modest performance decline during the prediction of 0.024 In F1-Score with a larger decline of 0.086 for clustering (**Supplementary Fig. 14a**).

Extract of Supplementary Figure 14 | Extensions of mvTCR. (...) b, Clustering of mvTCR with TCR, transcriptome, and surface protein markers compared to the baseline model on the SARS-CoV-2 dataset. (...)

- While this can sufficient for some use cases, more sophisticated modeling of missing information might further improve the model's representation as we now suggest as future work in the discussion section (**lines 449-451**: “While training mvTCR on partially missing data might be sufficient for prediction (Supplementary Fig. 14a), novel mosaicing techniques [59-61] could further overcome the decrease in information content.”)

C2.3: How does the model handle the inclusion of a supervised component to predict epitope specificity? Can this enhance the model's predictive capabilities?

R2.3: We thank the reviewer for this idea. To test this, we adopted mvTCR by a supervised classification head predicting antigen specificity from the multimodal representation.

- The classification component follows a standard semi-supervised learning scheme now explained in the methods section (**lines 574-578**: “Supervised Classification Head: The architecture consists of fully connected layers, batch-normalization, ReLU activation, and dropout followed by a final linear layer with softmax activation. The distribution over all specificity labels is predicted from the mean of the joint latent representation μ . The network is trained simultaneously with the VAE to minimize an additional cross-entropy loss, while the weighting of this loss is a tunable hyperparameter.”)
- We observed that the resulting representation improves clustering and partial kNN prediction (**lines 223-228**: “To further enhance the representation, we adapted mvTCR to semi-supervised learning introducing a supervised classification head to predict antigen specificity from μ^i_{joint} . While direct prediction fell short, the kNN classifier on the resulting representation surpassed the base model on three, and clustering on all six datasets (Supplementary Fig. 7) with an average improvement of 0.12 in NMI. These results might further be improved by methods tackling the trade-off between the unsupervised and supervised learning objectives.”, **Supplementary Figure 7b**)

b

Model	F1-Score							NMI						
	10x Full	D1	D2	D3	D4	Minervina	Total	10x Full	D1	D2	D3	D4	Minervina	Total
mvTCR	0.80	0.82	0.84	0.89	0.78	0.79	0.82	0.50	0.58	0.41	0.00	0.52	0.44	0.41
Sup-Repr	0.84	0.63	0.86	0.89	0.77	0.83	0.80	0.61	0.63	0.54	0.05	0.63	0.72	0.53
Sup-Class	0.51	0.43	0.62	0.50	0.28	0.78	0.52							

Supplementary Figure 7 |Additional specificity benchmarks. Capturing of pMHC specificity by atlas-query prediction (weighted F1-Score) and clustering (Normalized Mutual Information) on the 10x Genomics dataset for all donors, donors 1-4 separately, and the Minervina dataset. Each score represents the average over n=5 random splits. (...) **b**, Comparison of the base model to the results directly obtained with the supervised classification head (Sup-Class) and its representation (Sup-Repr).

- Consequently, we removed the corresponding section from the future work description. (**lines 455-457**: “Adding a supervised component to the model to predict epitope specificity could further guide the network's training and improve the multimodal representation.”)

C2.4: The paper discusses the limitations of adjusting the contribution of each modality. Are there any strategies or future directions for automating this process and making it more adaptable to different analysis objectives?

R2.4: We agree with the reviewer that adjusting the weighting between both modalities is a critical point. Therefore, we proposed two ways to optimize the weighting. When a relevant annotation is provided, the model can be trained to predict it best (see 10x and Minervina dataset). If it remains open which properties are relevant during analysis, it might be necessary to train the model several times using the preservation of e.g. clonotype and cell type to guide the training (see Fischer, TIL, Minervina dataset).

- Unfortunately, the weighting between both modalities is not only dataset but also analysis-dependent. E.g. while for some analysis conserving antigen-specificity might be of greater relevance, one might also want to analyze the same dataset with a focus on other properties such as disease severity or outcome. We now explain this in more detail in our manuscript. (**lines 470-473:** “Even though an automated selection of network parameters could partially prevent retraining, the desired contribution of each modality is dependent on the study and might vary across or even within one dataset depending on the analysis objective.”)

C2.5: Can mvTCR be applied to other immune cell types, such as B cells, and how might the modeling of somatic hypermutations be addressed in such cases?

R2.5: The reviewer raises an interesting point, as the application of mvTCR to B cells would increase its applicability to novel research questions. From a technical point of view, mvTCR can directly be applied to scGEX+scBCR datasets as well. However, benchmarking its performance in this setting is problematic as specificity annotation of large B-cell datasets of several 10k cells is less established than for T cells.

- To investigate whether somatic hypermutation would be captured in the joint model's TCR embedding, we analyzed clusters of TCR clones with single-linkage Hamming distance of 1 in the beta chain to mimic lineage trees in the SARS-CoV-2 dataset (**lines 789-792:** “To investigate whether somatic hypermutation is potentially captured by mvTCR we defined single-linkage clusters of TCR clones with a Hamming distance in their CDR3beta-chain. The average distances were calculated on the joint model's TCR space (z^i_{TCR}) within the cluster or a random clone selection of the same size.”).
- As expected, we observed that TCR clones within a Hamming cluster lie significantly closer in the mvTCR representation indicating that BCR lineages would potentially directly be captured by our approach (**lines 464-468:** “Tests on the SARS-CoV-2 dataset showed that TCR clusters with single amino acid mutations in their CDR3beta region have significantly lower distance in the TCR representation of our joint model (Supplementary Fig. 14c). While this indicates that somatic hypermutations of BCRs might be inherently captured by mvTCR, an in-depth benchmark must be conducted upon the availability of large-scale B cell datasets with specificity annotation.”, **Supplementary Fig. 14c**).

Extract of Supplementary Figure 14 | Extensions of mvTCR. (...) c, Average distances within clusters of TCR clones with single amino-acid mutations in the CDR3 β -chain compared to random clusters in the SARS-CoV-2 dataset (p-values: * <0.05 , ** <0.01 , *** <0.001).

C2.6: In real-world applications, how practical is the implementation of mvTCR in terms of computational resources and time required for training and analysis?

R2.6: We understand the reviewer's concern as the purpose of mvTCR is the analysis of relevant immunological studies, and its usability is therefore of great importance. To show the scalability, we conducted a timing analysis (Fig. 5c) which resulted in training times of under 10 minutes for 100k cells. Better performance was achieved when conducting a hyperparameter optimization for 48 or 96 GPU hours.

- For transparency, we added the computational resources used for conducting the training of the presented models to the manuscript (**lines 650-653**: "All experiments except the timing analysis were either conducted on a single GPU machine of 32GB of memory or paralyzed to train 4 models simultaneously on a node containing 4 GPUs and 512GB of memory."). Note, that the amount of memory was mainly determined by the specification of our computational cluster instead of actual requirements to train the models.
- To test the minimal requirements we conducted the HPO on the full 10x dataset (~61k cells) and observed that 8GBs of memory is sufficient. As this experiment run was not included in the manuscript, we added these requirements to the **GitHub description** of our tool rather than the main text.
- To ensure usability, we updated the tutorials on processing and training mvTCR in our **GitHub repository** and made the tool conveniently installable via pip. Additionally, we refactored the API of our package, so that a single-cell dataset can be preprocessed into the appropriate format and trained with mvTCR with only two function calls.
- The embedded representation is fully interoperable with standard single-cell analysis methods for GEX such as visualization and clustering through the Scanpy / AnnData data format as added to the discussion (**lines 437-438**: "mvTCR's representation seamlessly integrates into standard single-cell analysis workflows over the Scanpy-format [55].").

C2.7: Are there any potential applications for mvTCR in clinical settings, such as disease diagnosis or personalized treatment planning?

R2.7: We agree with the reviewer that GEX and TCR information could provide information in clinical settings. Zaslavsky et al. (preprint 2023) showed initial steps for classifying between four disease states (COVID-19, HIV, Lupus, and Healthy) based on the patient's immune receptor repertoires (TCRs and BCRs). Adding the GEX profile of the cells could further improve predictive performance. In this line, our analysis of the SARS-CoV-2 dataset showed the separation of cells from donors with different severity classes. However, we believe that due to the high cost and time compared to other tests, single-cell sequencing will not have a major impact on diagnostic or treatment planning. We rather envision mvTCR to aid in the research, which impacts diagnosis and treatment planning by fostering our knowledge of diseases and T-cell biology.

C2.8: Can mvTCR be applied to datasets from diverse species, or are there limitations regarding the type of organisms for which it can be effectively utilized?

R2.8: We envision that the integration of TCR and transcriptome will discover currently hidden mechanisms in T cell biology. As the tool should therefore as broadly applicable as possible, we do not utilize any information specific to individual species.

- Therefore, there are no limits concerning species as long as paired single-cell GEX and TCR sequences are available as their format does not differ between species. We added this clarification in the discussion section (**lines 426-427**: “As the model solely uses the transcriptome counts and the TCR sequences, it is theoretically applicable across species.”).
- In practice, the Cell Ranger software accompanying the widely-used commercial 10x Genomics kits only provides reference genomes of GEX and VDJ (TCR and BCR) for human and murine samples. Hence, the majority of datasets on which mvTCR will be applied stem from these two species. As this is a general limitation of single-cell sequencing rather than mvTCR, we did not add this information to our manuscript.

C2.9: How does mvTCR compare with other state-of-the-art methods for integrating multimodal single-cell data in terms of performance and computational efficiency?

R2.9: In our work, it was of great importance to compare against state-of-the-art multimodal models to ensure the performance of mvTCR. However, most other single-cell integration tools were developed for modalities represented as count matrices and, hence, can not be applied to TCR sequences.

- In the revised manuscript, we clarified this in the introduction (**lines 58-60** “While integration tools exist for other modality combinations such as transcriptome and surface protein counts, these methods are not adapted to the TCR protein sequences.”) and discussion (**lines 423-424** “However, standard multimodal methods cannot be applied as they are not adapted model the TCR sequence.”) section.

Throughout the manuscript, we conducted an extensive evaluation against two tools specifically developed for joint analysis of TCR and Gene expression data.

- mvTCR surpasses tessa's (Zhang et al., 2021) TCR clone embedding regarding specificity prediction (Fig. 3c) and conservation of cell types (Supplementary Fig. 9b).

- As CoNGA (Schattgen et al., 2022) rather selects small groups of cells with similar TCR and GEX and therefore does not provide an embedding or cluster annotation for all cells, the comparison was not directly applicable.
- However, we analyzed the computational efficiency measured by the time required to apply both methods (Fig. 5c) and discovered that mvTCR trains considerably faster across all dataset sizes. Further, in contrast to mvTCR, both SOTA methods could not be applied to large-scale datasets as they exceed the 256GB of memory at 30k (tessa) and 100k cells (CoNGA).

REVIEWER COMMENTS

Reviewer #1 (Remarks to the Author):

We would like to thank the authors for a thorough response to all our questions. Indeed, the clarity of the text in both the manuscript and the online software manual has increased a lot compared to the initial version.

We thank the authors for coloring UMAP plots by donor in the updated Figure 2a. However, this now raises additional questions:

1) The RNA-only model in Figure 2a shows clustering heavily influenced by the Donor ID (e.g. donor 2 is clustered separately from other donors). Would you see the same clustering result (i.e. donor 2 phenotypes are very different from other cells in the dataset) if you analyzed your scRNA-Seq data with standard unimodal data integration tools (e.g. Harmony or Seurat Integration)? Unfortunately, current plots suggest the absence of rational integration, which essentially means prohibiting any informative analysis based on the identification of cell clusters with this or that functionality present across patients, even at this basic scRNA-Seq level.

2) Another possible way to address the same question would be coloring various T cell markers (e.g. GZMA, GZMK, GZMB, GZMH, CD4, CD8A, CD8B, TBX21, SELL, PDCD1, NKG7) on your UMAP (Figure 2a, RNA-only model). This might help to assess whether the cells derived from donor2 indeed are so different phenotypically from the other cells, that this phenotypical difference makes them cluster separately. Maybe other genes make these cells phenotypically distinct?

For now, we suspect that the RNA-only-based model produces outputs influenced heavily by batch effects. This might be that the already existing tools outperform the RNA-only model provided by the authors (in terms of the cell clustering). The manuscript is partially built on the comparison of the mvTCR performance over the RNA-only model performance (e.g. Figures 2a, 3a, 3d). However, if the RNA-only model itself works poorly (fails to cluster cells of the same phenotype originating from the different donors correctly), then the idea of this comparison (RNA-only versus mvTCR) might be unfair, and the rationality of the whole mvTCR analysis remains questionable.

The second reviewer raised an important topic in the discussion:

"C2.9: How does mvTCR compare with other state-of-the-art methods for integrating multimodal single-cell data in terms of performance and computational efficiency?"

R2.9: In our work, it was of great importance to compare against state-of-the-art multimodal models to ensure the performance of mvTCR. However, most other single-cell integration tools were developed for modalities represented as count matrices and, hence, can not be applied to TCR sequences."

However, it seems that a deeper investigation is needed to compare the proposed by the authors RNA-only solution with the existing tools aimed to work with unimodal data. And on top of that, a fair mvTCR performance assessment might be possible.

Reviewer #2 (Remarks to the Author):

I am happy how the comments were addressed.

In the following, the editor's and reviewers' comments appear in **blue**, our point-by-point answers appear in **black**, and passages and figures we have added or altered in the manuscript are highlighted in **red** (first revision) and **blue** (second revision) in the paper. Short additions are directly added to the point-to-point response, while longer sections are referenced by their line numbers in the revised manuscript.

REVIEWER COMMENTS

Reviewer #1 (Remarks to the Author):

We would like to thank the authors for a thorough response to all our questions. Indeed, the clarity of the text in both the manuscript and the online software manual has increased a lot compared to the initial version.

We thank the authors for coloring UMAP plots by donor in the updated Figure 2a. However, this now raises additional questions:

1) The RNA-only model in Figure 2a shows clustering heavily influenced by the Donor ID (e.g. donor 2 is clustered separately from other donors). Would you see the same clustering result (i.e. donor 2 phenotypes are very different from other cells in the dataset) if you analyzed your scRNA-Seq data with standard unimodal data integration tools (e.g. Harmony or Seurat Integration)? Unfortunately, current plots suggest the absence of rational integration, which essentially means prohibiting any informative analysis based on the identification of cell clusters with this or that functionality present across patients, even at this basic scRNA-Seq level.

2) Another possible way to address the same question would be coloring various T cell markers (e.g. GZMA, GZMK, GZMB, GZMH, CD4, CD8A, CD8B, TBX21, SELL, PDCD1, NKG7) on your UMAP (Figure 2a, RNA-only model). This might help to assess whether the cells derived from donor2 indeed are so different phenotypically from the other cells, that this phenotypical difference makes them cluster separately. Maybe other genes make these cells phenotypically distinct?

For now, we suspect that the RNA-only-based model produces outputs influenced heavily by batch effects. This might be that the already existing tools outperform the RNA-only model provided by the authors (in terms of the cell clustering). The manuscript is partially built on the comparison of the mvTCR performance over the RNA-only model performance (e.g. Figures 2a, 3a, 3d). However, if the RNA-only model itself works poorly (fails to cluster cells of the same phenotype originating from the different donors correctly), then the idea of this comparison (RNA-only versus mvTCR) might be unfair, and the rationality of the whole mvTCR analysis remains questionable.

The second reviewer raised an important topic in the discussion:

“C2.9: How does mvTCR compare with other state-of-the-art methods for integrating multimodal single-cell data in terms of performance and computational efficiency?

R2.9: In our work, it was of great importance to compare against state-of-the-art multimodal models to ensure the performance of mvTCR. However, most other single-cell integration tools were developed for modalities represented as count matrices and, hence, can not be applied to TCR sequences.”

However, it seems that a deeper investigation is needed to compare the proposed by the authors RNA-only solution with the existing tools aimed to work with unimodal data. And on top of that, a fair mvTCR performance assessment might be possible.

We thank the reviewer for this suggestion as batch effects are a typical problem in many datasets to which mvTCR could be applied. So far, we considered the batch effects as a dataset limitation equally challenging baseline and mvTCR model in our benchmarks. As suggested by the reviewer, we further investigated the cause of the batch effect and evaluated whether the integration tool Harmony provides a better embedding than our proposed uni-modal baseline. In detail, we conducted the following evaluation:

- As suggested by the reviewer, we investigated the genes causing the distinctive clustering between the donors. There is no clear separation pattern in the common T cell marker genes provided by the reviewer. CD4, CD8A, and CD8B were not shown here, as the dataset was sorted for CD8⁺ cells and, therefore, these genes were not contained in the 5000 most variable genes. However, several genes reported to be a signature of *ex vivo* activation after PBMC processing at room temperature (*FOS*, *JUN*, *JUNB*, *ZFP36*, *DUSP1*, *DDIT4*, *NFKBIA*, *CXCR4*) [Andriamboavonjy et al., 2023] were contained in the top 20 differentially expressed genes of Donor 2 indicating the reason for separating the donor from the remaining dataset. (**lines 133-139**: “As *GLCTLVAML* and *RAKFKQLL* were bound almost exclusively by Donor 2 (**Supplementary Fig. 4**), their clusters could be partially identified by shifts in the transcriptomic profile, which shows a clear separation from the remaining dataset. While common T cell markers were distributed across all donors (**Supplementary Fig. 5a**), Donor 2 differentially expressed several *ex vivo* activation signature genes such as *FOS*, *DUSP1*, *JUN*, and *NFKBIA* [27] (**Supplementary Figure 5b-d**).”)

Supplementary Figure 5 | Donor effects in the 10x dataset. UMAP visualization of the RNA model colored by selected T cell marker genes (a), differentially expressed *ex vivo* activation signature genes (b), and donor ID (c). d, Differentially expressed genes of Donor 2 compared to the remaining dataset.

- We performed Harmony-correction based on the donor ID resulting in an integrated RNA embedding. We repeated the analysis in Figure 2 directly on the corrected Principal Components (PCs) provided by Harmony. We found that the PCs from the correction did not improve but harm clustering for specificity indicated by a lower NMI of 0.212 (RNA: 0.46, mvTCR: 0.535), which may be caused by correcting the proliferation imprint of Donor 2 (**lines 768-771**: “For analysis on the batch-corrected dataset, we applied Harmony [28] at its default parameters. Models trained on the batch-corrected dataset directly received the PCs provided by Harmony as the transcriptome input.”, **lines 139-141**: “However, correcting for donor effects using Harmony integration (**Supplementary Figure 6a-b**) decreased the clustering performance to an NMI of 0.212 at $n=14$ clusters.”)

Supplementary Figure 6 | Harmony correction of the 10x dataset. UMAP visualization of the 10x dataset using Harmony [28] for batch correction colored by donor, cell type, and specificity (**a**) as well as selected T cell marker genes (**b**).

- Additionally, we repeated the benchmark of Figure 3a for the pooled 10x dataset with batch correction on five splits. For this, we utilized the corrected PCs, a VAE model trained on these PCs, and the mvTCR model trained on these PCs in addition to the TCR sequence. (lines 227-234: “Following, we utilized the batch-corrected dataset to ensure that the separation of Donor 2 does not impair the performance in the transcriptome baseline for the pooled 10x dataset. Besides the PCs provided by Harmony [28], we evaluated models trained on the PCs (Harmony-RNA) and additionally the TCR sequences (Harmony-mvTCR). While for the PCs the performance decreased considerably by an F1-Score of 0.239 and an NMI of 0.163 over the RNA model, the models trained on batch-corrected data performed similarly to models trained on the corresponding uncorrected dataset (**Supplementary Figure 10**).

Supplementary Figure 10 | Benchmark using Harmony batch correction. Specificity prediction (F1-Score) and clustering performance (NMI) on the pooled 10x dataset using the Principal Components (PCs) provided by Harmony [28], a VAE model trained on these PCs (Harmony-RNA), a VAE model trained on these PCs and TCR sequences (Harmony-mvTCR), and the two original models introduced earlier (RNA, mvTCR). The bars represent the average metric score, while the error bars indicate the 95% confidence interval.

- We want to highlight, that the remaining specificity benchmark on the 10x dataset (**Figure 3a**) was conducted for each donor individually and, therefore, is not compromised by potential batch effects, where the mvTCR model clearly outperforms the RNA baseline for Donor 1, 2, and 4.

In summary, we show that the separation of Donor 2 from the remaining dataset largely stems from a cell processing induced ex vivo activation signature. Batch correction did not show to be advantageous for predicting and clustering antigen specificity in this dataset. Taking this and the remaining benchmarks into account, we conclude that our initial premise holds, that integrating transcriptome and TCR sequences yields a better representation with respect to antigen specificity than unimodal models.

Reviewer #2 (Remarks to the Author):

I am happy how the comments were addressed.

We thank the reviewer for the valuable discussions that considerably improved our manuscript.

REVIEWERS' COMMENTS

Reviewer #1 (Remarks to the Author):

I am satisfied with the answers and additional work performed, thank you.